# Iron-sulfur clusters are involved in post-translational arginylation

Verna Van[1], Janae B. Brown [1], Corin R. O'Shea [2], Hannah Rosenbach[3], Ijaz Mohamed[1], Nna-Emeka Ejimogu [1], Toan S. Bui[1], Veronika A. Szalai[4], Kelly N. Chacón [5], Ingrid Span [3], Fangliang Zhang[2,6] & Aaron T. Smith [1] ✉

Eukaryotic arginylation is an essential post-translational modification that modulates protein stability and regulates protein half-life. Arginylation is catalyzed by a family of enzymes known as the arginyl-tRNA transferases (ATE1s), which are conserved across the eukaryotic domain. Despite their conservation and importance, little is known regarding the structure, mechanism, and regulation of ATE1s. In this work, we show that ATE1s bind a previously undiscovered [Fe-S] cluster that is conserved across evolution. We characterize the nature of this [Fe-S] cluster and find that the presence of the [Fe-S] cluster in ATE1 is linked to its arginylation activity, both in vitro and in vivo, and the initiation of the yeast stress response. Importantly, the ATE1 [Fe-S] cluster is oxygen-sensitive, which could be a molecular mechanism of the N-degron pathway to sense oxidative stress. Taken together, our data provide the framework of a cluster-based paradigm of ATE1 regulatory control.

In eukaryotes, the half-life of a protein is largely regulated by the ubiquitin-proteasome system (UPS), which catalyzes the covalent attachment of the small protein ubiquitin (Ub) to Lys residues of polypeptides, providing a molecular flag for degradation via the 26 S proteasome[1–4]. The generation of degradation signals is dependent on the presence of an internal Lys residue (the site of the formation of the poly-Ub chain) and the identity of the N-terminal residue, the latter giving rise to the aptly named N-degron pathway (formerly known as the N-end rule pathway)[5–7]. A major component of this pathway is the three-level hierarchical Arg N-degron branch that requires downstream processing prior to Ub conjugation (Fig. 1a). As the first step of the Arg N-degron pathway, tertiary destabilizing N-terminal residues Asn, Gln, and Cys require modification by specific enzymes, dioxygen ($O_2$), or nitric oxide (NO), before further recognition. Specifically, Asn and Gln must be deamidated to Asp and Glu by the respective enzymes N-terminal asparagine amidohydrolase (NTAN) and N-terminal glutamine amidohydrolase (NTAQ)[8], whereas Cys must be oxidized to Cys-sulfinic or sulfonic acid[9]. These modifications yield secondary destabilizing residues, which are subsequently recognized by arginine transferases (known as ATE1s), essential enzymes of the Arg N-degron pathway that arginylate N-terminal residues in a tRNA-dependent manner to form primary destabilizing residues (Fig. 1)[10]. Finally, arginylated proteins may be recognized by E3 ubiquitin ligases, ubiquitinated, and transported to the proteasome for degradation (Fig. 1), or they may exhibit nondegradative alteration in function (Fig. 1)[6,11–13].

The fidelity of the post-translational arginylation, which plays both a degradative and nondegradative role, is critical for normal eukaryotic cellular function[4]. For example, key protein targets of the N-degron pathway are responsible for several essential physiological processes, from chromosomal segregation, to the stress response, to $Ca^{2+}$ homeostasis and cardiovascular development[7,9,14–21]. Important studies have additionally demonstrated a link between ATE1 function, the sensing of $O_2$ availability via plant cysteine dioxygenases (CDOs), and the oxidized Cys component of the Arg N-degron pathway[22–24]. Moreover, nondegradative post-translational arginylation has been observed for over a hundred intracellular targets and is arguably as

[1]Department of Chemistry and Biochemistry, University of Maryland, Baltimore County, Baltimore, MD 21250, USA. [2]Department of Molecular and Cellular Pharmacology, University of Miami, Miller School of Medicine, Miami, FL 33136, USA. [3]Institut für Physikalische Biologie, Heinrich-Heine-Universität Düsseldorf, 40225 Düsseldorf, Germany. [4]Physical Measurement Laboratory, National Institute of Standards and Technology, Gaithersburg, MD 20899, USA. [5]Department of Chemistry, Reed College, Portland, OR 97202, USA. [6]Sylvester Comprehensive Cancer Center, University of Miami Miller School of Medicine, Miami, FL 33136, USA. ✉e-mail: smitha@umbc.edu

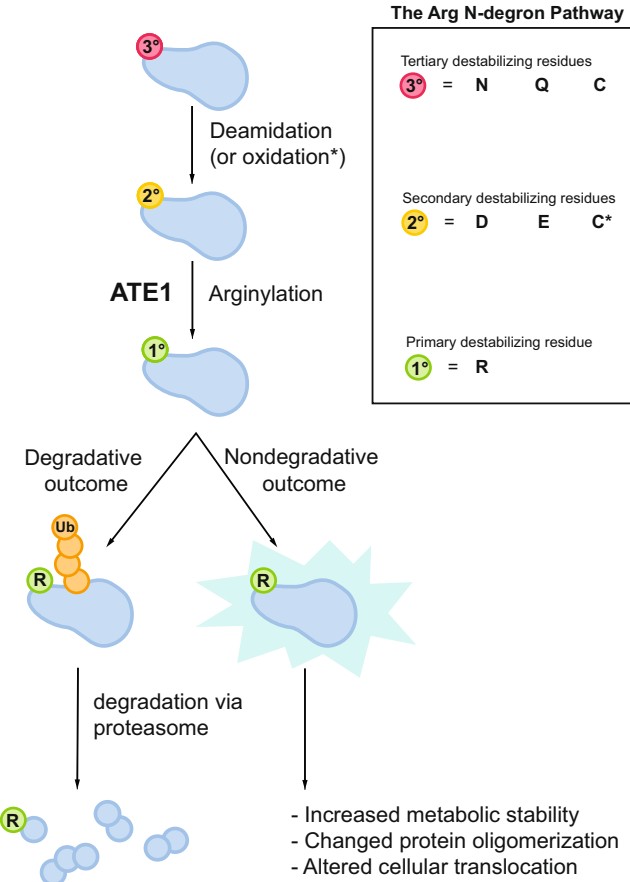

**The Arg N-degron Pathway**

Tertiary destabilizing residues

3° = N Q C

Secondary destabilizing residues

2° = D E C*

Primary destabilizing residue

1° = R

**Fig. 1 | Post-translational arginylation is an important degradative and nondegradative post-translational modification.** One function of arginylation is in the eukaryotic Arg N-degron pathway, a hierarchical pathway that controls the fate of many proteins in the cell. Tertiary destabilizing residues (red circles) are N-terminal Asn, Gln, and/or Cys residues that are exposed upon protein processing. These residues are modified via deamidation or oxidation to form the secondary destabilizing residues (yellow circles) Asp, Glu, and/or oxidized Cys (C*). Secondary destabilizing residues are recognized by arginine transferases (ATE1s), which enzymatically transfer Arg to the N-terminus of a polypeptide, resulting in the exposure of a primary destabilizing residue (green circle), that can be recognized by N-recognins, polyubiquitinated, and broken down in a proteasomal manner. Another important function is nondegradative arginylation, in which arginylated substrate polypeptides may have altered stability, oligomerization, cellular translocation, and even activity.

important as its role in the N-degron pathway[25,26]. Examples of key physiological processes controlled by nondegradative arginylation include cell migration, neurodegeneration, and even nucleotide biosynthesis[27–29]. However, these target proteins of ATE1-catalyzed arginylation are highly diverse and have varied N-terminal residues, requiring regulation at multiple levels for faithful arginylation.

Regulation of ATE1 may occur at the post-transcriptional as well as post-translational levels[4]. In some lower-order eukaryotes such as single-celled yeasts, only one (iso)form is thought to exist encoded along a single gene[30], while higher-order eukaryotes may have more than one *ate1* gene that can encode for several isoforms (Supplementary Fig. 1)[12,31,32]. Once translated, these different isoforms may differ by as little as *ca.* 5000 g/mol (5 kDa)[10,33]. However, as has been reported for *Mus musculus* (mouse) ATE1, these isoforms may exhibit important differences in tissue distribution, protein specificity, and even arginylation efficacy[10,34–36], demonstrating post-transcriptional regulation of the *ate1* gene. At the post-translational level, the ATE1-controlled Arg N-degron pathway may be regulated by the presence of small molecules

such as $O_2$, NO, and iron protoporphyrin IX (heme *b*)[18,23,37–39]. These results tantalizingly suggested that ATE1s could be heme-binding proteins that respond to small molecules in a manner similar to well-known gas-sensing proteins[40,41]. However, the precise molecular-level details of this regulatory process were essentially unknown.

In this work, we show that ATE1 unexpectedly binds an [Fe-S] cluster that regulates arginylation in vitro and in vivo. Using a mixture of spectroscopic, biochemical, and biophysical techniques, we characterize the as-isolated and reconstituted cluster-binding environments of yeast (*Saccharomyces cerevisiae*) ATE1 (*Sc*ATE1). These data demonstrate that *Sc*ATE1 binds a [4Fe-4S]$^{2+/+}$ cluster in its N-terminal domain that is oxygen-sensitive, redox-active, and capable of being converted to a [2Fe-2S]$^{2+}$ cluster under oxic conditions. Importantly, we reveal that the presence of the [Fe-S] cluster can affect the in vitro arginylation efficacy of *Sc*ATE1, and in vivo data demonstrate that the cluster-binding capacity of *Sc*ATE1 is necessary for a normal stress response in yeast. Lastly, using bioinformatics data, expression of mouse (*Mus musculus*) isoform ATE1-1 (*Mm*ATE1-1), reconstitution approaches, and spectroscopic characterization of *Mm*ATE1-1, we show that the presence of an [Fe-S] cluster is an evolutionarily conserved attribute of ATE1s. Based on these results, we propose an [Fe-S] cluster-based paradigm of ATE1 regulation that is likely operative across eukaryotes to control post-translational arginylation.

## Results

### *Sc*ATE1 unexpectedly binds an [Fe-S] cluster

We initially set out to characterize the heme-binding properties of yeast (*S. cerevisiae*) ATE1 (*Sc*ATE1), which exists as a single, soluble isoform. We cloned, expressed in *Escherichia coli*, and purified His-tagged *Sc*ATE1 (518 amino acids and 59,600 g/mol i.e., 59.6 kDa; Supplementary Fig. 2a, verified by Western blotting) to high purity and high oligomeric homogeneity (chiefly monomeric) as judged by SDS-PAGE and gel filtration (Supplementary Fig. 2b, c). Importantly, concentrated *Sc*ATE1 was a slightly brown color, indicating the presence of a putative cofactor, which we initially suspected was heme. However, metal analyses indicated very little iron (0.10 Fe ion ±0.06 Fe ion per polypeptide, *n* = 5), and the electronic absorption (EA) spectrum of this protein indicated no spectral features consistent with the presence of heme. Initially surmising that *Sc*ATE1 was a hemoprotein, and suspecting poor cofactor incorporation during expression, we supplemented our growth media with known heme *b* precursors such as ferric (Fe$^{3+}$) citrate and δ-aminolevulinic acid (ALA), which is a common strategy for increasing accumulation of heme-replete proteins[42,43]. Despite a modest color change in our cell pellets, and a slightly darker brown color of purified *Sc*ATE1 (Supplementary Fig. 2d, inset), the analyzed spectral features (Supplementary Fig. 2d) were wholly inconsistent with the presence of heme *b* in the purified enzyme[44,45]. They were, instead, remarkably similar to proteins containing iron-sulfur ([Fe-S]) clusters[46,47].

Presuming that *Sc*ATE1 might actually be an [Fe-S] cluster-binding protein, we modified our expression and purification protocol to facilitate cluster incorporation in several ways: media supplementation with ferric citrate and L-Cys, as well as co-expression with a high-copy plasmid encoding the *E. coli* iron-sulfur cluster (ISC) biosynthesis machinery (pACYC184*iscS-fdx*)[48]. Although expression of *Sc*ATE1 in the presence of only ferric citrate and L-Cys dramatically changed the color of the *E. coli* culture and appeared to increase substantively the spectral features associated with the cluster, reproducibility was problematic. We finally achieved high reproducibility of *Sc*ATE1 expression in the presence of pACYC184*iscS-fdx* (constitutively active at low levels) in media supplemented with ferric citrate and L-Cys. After expression under these conditions and oxic purification, the darkly colored *Sc*ATE1 (Fig. 2a), which maintained its monomeric oligomerization, exhibited an electronic absorption (EA) spectrum diagnostic of a [2Fe-2S]$^{2+}$ cluster (Fig. 2b).

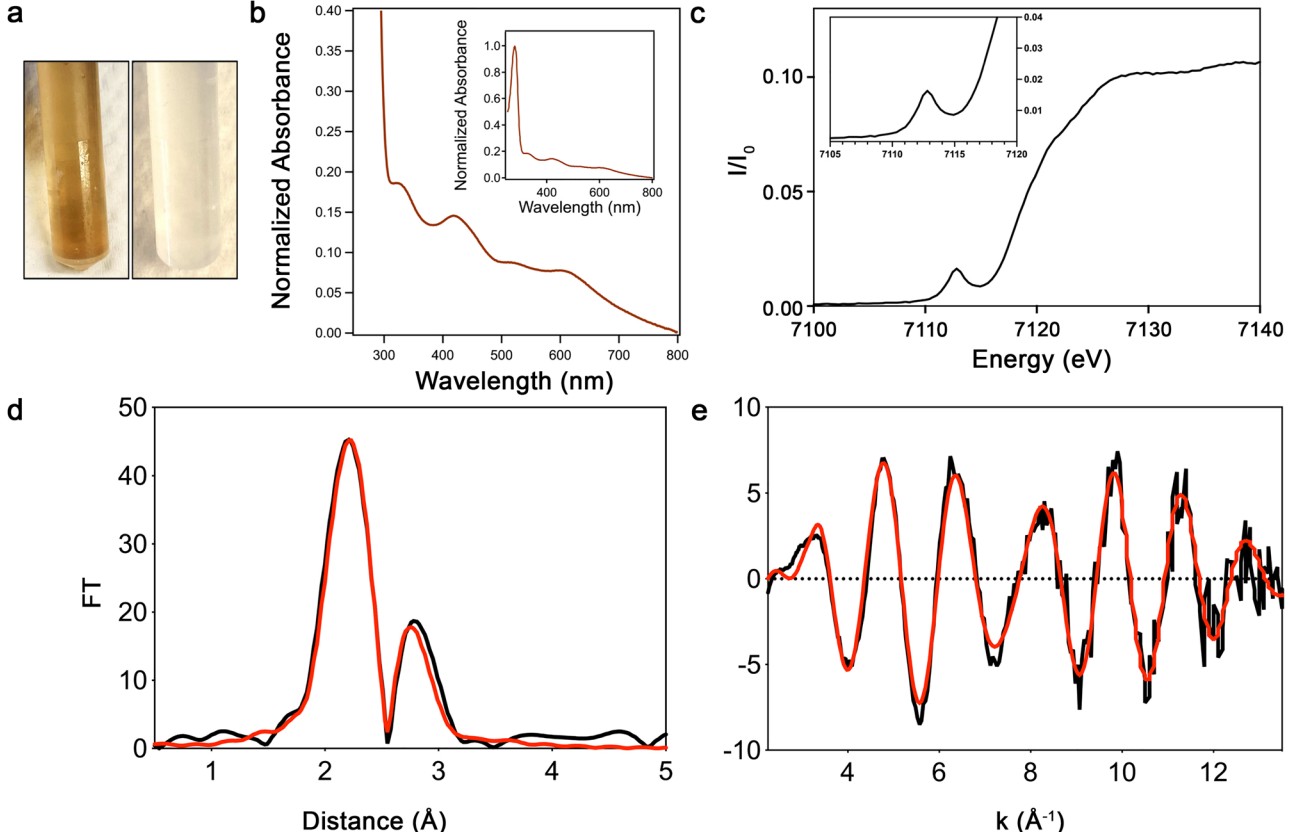

**Fig. 2 | *Sc*ATE1 binds a [2Fe-2S]²⁺ cluster when purified under oxic conditions.** **a** Photographs of plastic tubes containing purified *Sc*ATE1 (brown, left) and buffer (colorless, right) suggest the presence of an [Fe-S] cluster. **b** Features of the visible electronic absorption spectrum of *Sc*ATE1 isolated under oxic conditions indicate the presence of a [2Fe-2S]²⁺ cluster. Inset: expanded absorption spectrum showing both UV and visible regions. **c** The Fe XANES of *Sc*ATE1 isolated under oxic

conditions. Inset: expanded pre-edge feature of the $1s{\rightarrow}3d$ transition at 7112.5 eV, consistent with tetrahedral $Fe^{3+}$ sites. **d**, **e** EXAFS of *Sc*ATE1 isolated under oxic conditions. The Fourier transformed data (**d**) and raw EXAFS (**e**) are consistent with the presence of a [2Fe-2S]²⁺ cluster. The black traces represent the experimental data, while the red traces represent the EXCURVE-fitted data. 1 Å = 0.1 nm. Source data are provided as a Source Data file.

Characterization of *Sc*ATE1 expressed in the presence of the ISC machinery and isolated under an oxic atmosphere is most consistent with a single ferredoxin-like [2Fe-2S]²⁺ cluster bound to the polypeptide. Analysis of acid-labile iron using the ferrozine assay indicated the presence of (1.87 ± 0.40) mole eq. of iron per polypeptide ($n = 3$), and analysis of acid-labile sulfide using a modified method of Beinert's[49,50] indicated (2.11 ± 0.75) mole eq. of sulfide per polypeptide ($n = 3$), consistent with the [2Fe-2S] designation. The [2Fe-2S] assignment is confirmed based on the features of the EA spectrum of *Sc*ATE1 isolated under oxic conditions (Fig. 2b) and further spectroscopic characterization (*vide infra*). The *Sc*ATE1 EA spectrum of protein purified under oxic conditions displays characteristic absorption maxima in the near-UV and visible regions: *ca.* 330 nm (shoulder), 414 nm, *ca.* 475 nm (shoulder), and a broad absorbance envelope from 500 to 700 nm, similar to proteins bearing [2Fe-2S]²⁺ ferredoxin-like clusters[48,51–54]. This absorbance spectrum is accompanied by a weak circular dichroism (CD) spectrum in the visible region (Supplementary Fig. 3), indicating the presence of the cluster within the polypeptide.

To characterize the [Fe-S] cluster further, we analyzed *Sc*ATE1 expressed in the presence of the ISC machinery and isolated under oxic conditions using X-ray absorption spectroscopy (XAS). The X-ray absorption near-edge structure (XANES) is shown in Fig. 2c and displays a distinct pre-edge feature at 7112.5 eV, which is characteristic of tetrahedral $Fe^{3+}$ centers. Further evidence for this oxidation state assignment is the first inflection point of the edge at ≈7120 eV, common for $Fe^{3+}$ centers (Fig. 2c). Simulations of the extended X-ray absorption fine structure (EXAFS) data of *Sc*ATE1 reveal only S-based environments as the nearest neighbor ligands with an Fe-S distance of

2.25 Å (1 Å = 0.1 nm; Fig. 2d, e and Table 1). Furthermore, there is no indication of O/N-nearest neighbor ligands, nor any indication of Fe-C scattering, precluding the involvement of the (His)₆ tag in iron binding, and ruling out the Rieske- (2 Cys/2 His) or mitoNEET-type (3 Cys/1 His) [2Fe-2S] cluster types. Long-range scattering interactions representing only a single Fe-Fe vector are observed and fitted to a distance of 2.71 Å (Fig. 2d and Table 1), strongly supporting the [2Fe-2S] designation. Altogether, the metal analyses, EA, and XAS data clearly point to the presence of a single ferredoxin-like [2Fe-2S]²⁺ cluster in oxically-purified *Sc*ATE1. To further corroborate the oxidation state of the [2Fe-2S] cluster, we characterized *Sc*ATE1 expressed in the presence of ISC and isolated under oxic conditions using continuous-wave (CW) X-band electron paramagnetic spectroscopy (EPR). Consistent with our oxidation state assignment of the [2Fe-2S]²⁺ cluster based on XAS, we observe an EPR-silent spectrum under these conditions (Supplementary Fig. 4a; manifested by two antiferromagnetically-coupled $Fe^{3+}$ ions). We then added 1 mmol L⁻¹ (final concentration) of either sodium ascorbate or sodium dithionite under strict anoxic conditions to the protein to reduce the cluster to the [2Fe-2S]⁺ state. In the presence of sodium ascorbate, we observed no change in the EA spectrum or the EPR spectrum. In contrast, in the presence of sodium dithionite, the EA spectrum was slowly altered and effectively bleached slowly over a course of *ca.* 60 min (Supplementary Fig. 4b). We then used both the ascorbate- and dithionite-reacted species in an attempt to find an EPR signal of the $S = ½$ [2Fe-2S]⁺ state of the *Sc*ATE1 cluster. Despite exhaustive attempts across multiple microwave powers and multiple cryogenic temperatures (4 to 100 K), we were unable to detect a diagnostic signal of any reduced cluster. Given the slow reaction to

**Table 1 | Fits obtained for the Fe K-EXAFS ScATE1 purified under oxic conditions and containing a [2Fe-2S] cluster, and ScATE1 reconstituted under anoxic conditions and containing a [4Fe-4S] cluster**

| Sample | Fit index[a] | Fe-S | | | Fe-Fe | | | Fe-Fe | | | $E_o^e$ (eV) |
|---|---|---|---|---|---|---|---|---|---|---|---|
| | | No[b] | R[c] (Å)[f] | DW[d] (Å$^2$) | No | R (Å) | DW (Å$^2$) | No | R (Å) | DW (Å$^2$) | |
| ScATE1 purified oxically | 0.45 | 3.5 | 2.25 | 0.009 | 1 | 2.71 | 0.007 | | | | −5.40 |
| ScATE1 reconstituted anoxically | 0.41 | 4 | 2.25 | 0.010 | 2 | 2.69 | 0.008 | 1 | 2.49 | 0.012 | −0.02 |

Fits were determined by curve fitting using the program EXCURVE (version 9.2). Certain commercial entities, equipment or materials may be identified in this document to describe an experimental procedure or concept adequately. Such identification is not intended to imply recommendation or endorsement by the National Institute of Standards and Technology, nor is it intended to imply that the entities, materials or equipment are necessarily the best available for the purpose.
[a]The least-squares fitting parameter (see Methods).
[b]Coordination number.
[c]Bond length.
[d]Debye–Waller factor.
[e]Photoelectron energy threshold.
[f]1 Å = 0.1 nm.

even the high-potential reductant sodium dithionite, and the lack of a distinct $S = ½$ [2Fe-2S]$^+$ EPR species, we surmise that reduction of the [2Fe-2S]$^{2+}$ form of ScATE1 produces an unstable species that results in cluster decomposition. This hypothesis is supported by the observation that anoxic desalting of the reduced protein and subsequent exposure of ScATE1 to oxygen did not restore the diagnostic EA signal of the [2Fe-2S]$^{2+}$ cluster.

## ScATE1 binds higher-order [Fe-S] clusters under anoxic conditions

Given that many [Fe-S] clusters can be sensitive to the presence of oxygen, and that the ScATE1 cluster appeared to be redox unstable, we then considered whether ScATE1 might be capable of binding a higher-order cluster (e.g., [3Fe-4S] or [4Fe-4S]), which we tested by reconstitution under anoxic conditions. When we titrated a maximum of 10 mole eq. of Fe$^{3+}$ and 10 mole eq. of S$^{2-}$ into apo ScATE1 over a several-hour period, we noted heavy precipitation of excess iron sulfide. After gel filtration, the absorption spectrum of ScATE1 reconstituted with excess cluster components suggested the presence of a distinctly different [Fe-S] cluster than observed under oxic conditions, although there was a broad background absorption indicative of residual spuriously-bound iron sulfide. To circumvent this problem, we titrated apo ScATE1 with a maximum 4 mole eq. of Fe$^{3+}$ and 4 mole eq. of S$^{2-}$ over a period of ≈2 h, which dramatically reduced the amount of precipitated iron sulfide and spurious binding. After filtration and buffer exchanging, we measured a substantive increased in the iron- and sulfide:polypeptide stoichiometries: 4.15 Fe ± 0.60 Fe; 3.00 S$^{2-}$ ± 0.27 S$^{2-}$; $n = 3$. The EA spectrum of ScATE1 reconstituted under anoxic conditions was most consistent with the presence of a [4Fe-4S] cluster (Fig. 3a), as only one major absorption band was present and centered at ca. 405 nm, common for [4Fe-4S]$^{2+}$ clusters[54–56]. We further confirmed this assignment using XAS and paramagnetic NMR spectroscopies. XAS again showed a first inflection point of the edge at ≈7120 eV and an intense pre-edge feature consistent with tetrahedral Fe centers (Supplementary Fig. 5). The EXAFS data (Fig. 3b, c) were fitted with all S-based environments as the nearest neighbor ligands with an Fe-S distance of 2.25 Å (Fig. 3b, c and Table 1). In addition, the best long-range scattering interactions required multiple Fe-Fe vectors (2 longer and 1 shorter) that were fitted to distances of 2.69 and 2.49 Å, respectively (Fig. 3b, c and Table 1), wholly consistent with a [4Fe-4S] cubane-like cluster. The EPR spectrum of ScATE1 reconstituted under anoxic conditions was again EPR-silent, indicating the [4Fe-4S] cluster was likely in the 2+ oxidation state, as is the common oxidized state of [4Fe-4S] clusters. Finally, to confirm the cluster's [4Fe-4S]$^{2+}$ identity, we took advantage of paramagnetic nuclear magnetic resonance (NMR) spectroscopy, which is capable of fingerprinting the unique compositions and oxidations states of [Fe-S] clusters due to paramagnetic Cys βCH$_2$ hyperfine shifts (Fig. 3d)[57,58]. The 1D $^1$H NMR spectrum of proteins

containing [4Fe-4S]$^{2+}$ clusters demonstrates 3 distinct and characteristic $^1$H shifts that are weak, located in the 10 ppm to 20 ppm region, and characteristic of this cluster composition[57,58]. Consistent with these established characteristics, we observe three weak hyperfine shifts ca. 11 ppm, 15 ppm, and 18 ppm in the 1D $^1$H NMR spectrum of ScATE1 reconstituted under anoxic conditions (Fig. 3d). When considered in combination with our other spectral data, it becomes clear that ScATE1 is capable of binding a [4Fe-4S]$^{2+}$ cluster.

Next, to test whether the [4Fe-4S] or the [2Fe-2S] cluster were present in vivo, we sought to purify ScATE1 under completely anoxic conditions after expression in anaerobic E. coli. To do so, we co-transformed ScATE1 pET-21a(+) into E. coli cells in which the chromosomal copy of the transcriptional repressor protein of the isc operon was deleted (ΔiscR). Due to high constitutive levels of expressed ISC machinery, this approach is a proven method to improve in vivo [Fe-S] cluster insertion under anaerobic growth conditions[48,59]. Bacteria were aerobically cultivated, then transferred to a glove box to induce expression. Cells were subsequently harvested under an inert atmosphere (Ar$_{(g)}$), lysed, and subjected to anoxic purification. The yield of anoxically purified ScATE1 was significantly diminished compared to oxic purification but exhibited good final purity (Supplementary Fig. 6, inset). Importantly, the EA spectrum of the ScATE1 expressed and purified under anoxic conditions exhibited discrete absorption maxima at ca. 330 nm and ca. 405 nm, which closely resembles a mixture of [3Fe-4S] and [4Fe-4S] clusters (Supplementary Fig. 6)[54–56,60,61]. We surmise that this mixture is the result of partial or incomplete cluster transfer to ScATE1 during anaerobic cell cultivation and heterologous protein production, or due to extreme O$_2$ sensitivity in the presence of a minute amount of residual O$_2$ during anoxic purification. To confirm these assignments, we collected the CW X-band EPR spectra of ScATE1 expressed and purified under anoxic conditions, both with and without the presence of sodium dithionite. In the absence of dithionite, we observed two major signals after subtraction of the background spectrum (i.e., buffer only): one sharp signal at $g ≈ 4.30$ consistent with adventitious or misloaded Fe$^{3+}$, and a second axial signal with $g_{||} ≈ 2.00$ and $g_⊥ ≈ 1.97$ consistent with a [3Fe-4S]$^+$ $S = ½$ species (Fig. 3e)[61,62]. In the presence of dithionite and after subtraction of the background spectrum (i.e., buffer with dithionite), we observed only a rhombic signal with $g_1 ≈ 2.03$, $g_2 ≈ 1.93$, and $g_3 ≈ 1.88$, consistent with a [4Fe-4S]$^+$ $S = ½$ species (Fig. 3e)[54,63,64]. Thus, when our anoxic reconstitution and purification results are taken together, it becomes evident that ScATE1 binds a higher-order cluster in the absence of oxygen and in vivo.

## The ScATE1 [Fe-S] cluster is oxygen-sensitive

Given the differences in the [Fe-S] cluster identity as a function of oxic vs. anoxic purification of ScATE1, we hypothesized that the ATE1 cluster might be oxygen-sensitive. We confirmed this hypothesis two different ways. First, after anoxic reconstitution and buffer exchanging

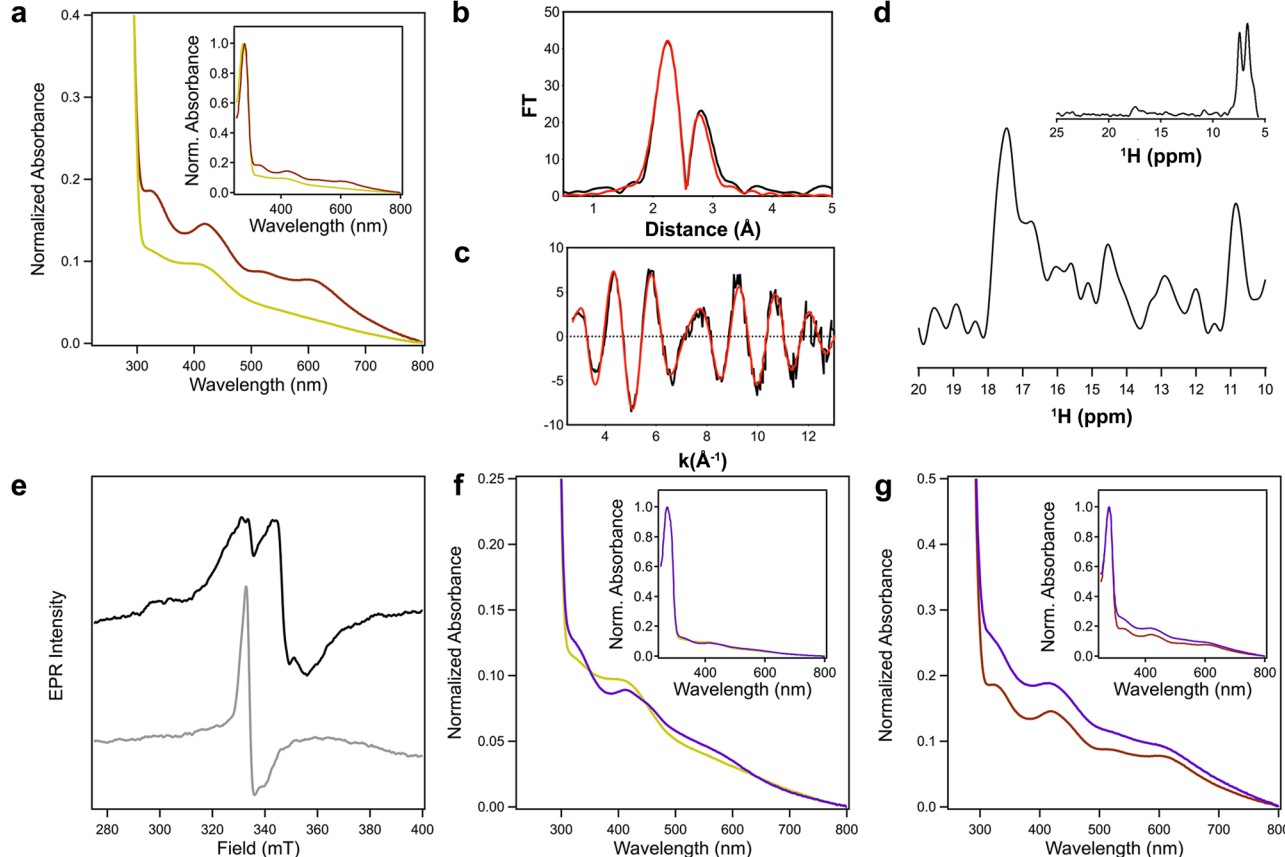

**Fig. 3 | Characterization and reactivity of the [4Fe-4S] cluster of ScATE1.**
**a** Reconstitution of ScATE1 under anoxic conditions yields an electronic absorption spectrum (yellow) consistent with the presence of a [4Fe-4S] cluster that is distinct from the [2Fe-2S] cluster obtained under oxic conditions (brown). **b**, **c** EXAFS of ScATE1 reconstituted under anoxic conditions. The Fourier transformed data (**b**) and raw EXAFS (**c**) are consistent with the presence of a [4Fe-4S] cluster. The black traces represent the experimental data, while the red traces represent the EXCURVE-fitted data. 1 Å = 0.1 nm. **d** Paramagnetic 1D $^1$H NMR data reveal three weak hyperfine shifts at 10.8 ppm, 14.5 ppm, and 17.5 ppm in the spectrum of ScATE1 reconstituted under anoxic conditions, consistent with a [4Fe-4S]$^{2+}$ cluster. Inset shows the entire 1D $^1$H NMR region from 5-25 ppm. **e** In the absence of sodium dithionite, an axial EPR spectrum is present that reveals the presence of some degraded [3Fe-4S]$^+$ cluster (gray), while a [4Fe-4S]$^+$ cluster is clearly observed in the dithionite-reduced EPR spectrum (black) of anoxically purified ScATE1. Conditions were the same for both samples: frequency = 9.38 GHz, temperature = 6 K, modulation amplitude = 0.5 mT, modulation frequency = 100 kHz, 1024 points, conversion time = 175.78 ms, microwave power = 1.9 mW, 16 scans. **f** Electronic absorption spectra reveal that the [4Fe-4S]$^{2+}$ cluster of ScATE1 reconstituted under anoxic conditions is reactive to $O_2$. The starting [4Fe-4S]$^{2+}$ spectrum prior to the $O_2$ reaction is yellow, and the ending [2Fe-2S]$^{2+}$ spectrum is purple. Inset: the full spectra are color-coded the same way. **g** Spectral comparisons reveal that the $O_2$-reacted form of the [4Fe-4S]$^{2+}$ cluster (purple) is identical to the [2Fe-2S]$^{2+}$ cluster present in ScATE1 isolated under oxic conditions (brown). Inset: the full spectra are color-coded the same way. Source data are provided as a Source Data file.

to remove excess iron and sulfide, exposure of the reconstituted [4Fe-4S]$^{2+}$ to ambient atmosphere resulted in rapid ($t_{1/2}$ « 5 min) conversion to a different species that strongly resembles a [2Fe-2S]$^{2+}$ cluster (Fig. 3f). Second, we performed the same experiment with ScATE1 that was produced and purified under anoxic conditions (mixture of [3Fe-4S]$^+$ and [4Fe-4S]$^{2+}$ clusters) and noted nearly identical behavior on a very similar timescale (Supplementary Fig. 7). In both cases, the final EA spectrum exhibited multiple peaks that were virtually super-imposable with that of EA spectrum of ScATE1 isolated under oxic conditions (Fig. 3g), pointing to the presence of a [2Fe-2S]$^{2+}$ cluster as a common species generated upon exposure of the higher-order cluster to $O_2$. These results demonstrate that the ScATE1 [Fe-S] cluster is oxygen-sensitive, which could be leveraged to sense oxidative stress within the cell in order to alter arginylation efficacy and/or substrate specificity.

## The ScATE1 [Fe-S] cluster is bound in the ATE1 N-terminal domain

To determine the binding location of the [Fe-S] cluster, we first used bioinformatics data on known ATE1 protein sequences to pinpoint Cys residues involved in cluster binding. Partial sequence alignments of

ATE1s from a continuum of organisms (yeast to human) reveal strong conservation of 4 Cys residues (numbered based on ScATE1): Cys$^{20}$/Cys$^{23}$ (forming a CxxC motif), and Cys$^{94}$/Cys$^{95}$ (forming a CC motif) (Supplementary Fig. 8). These Cys residues are clustered within the N-terminal region of the ATE1 polypeptide, which has an unknown function and is distinct from bacterial leucyl/phenylalanyl-(L/F)-tRNA protein transferases[4]. As four Cys residues are required to coordinate the [4Fe-4S] and [2Fe-2S] ferredoxin-like clusters, we hypothesized that the ATE1 N-terminal domain could have a function in [Fe-S] cluster binding.

To test this hypothesis, we first cloned the N-terminal domain of ScATE1 (residues 1–146) into a pET vector and attempted to express this domain in *E. coli*. Despite exhaustive efforts and expression testing using multiple cell lines and multiple media types, the tagged N-terminal domain failed to accumulate, suggesting this domain may be unstable on its own. Instead, we generated a construct of N-terminal ScATE1 fused to maltose-binding protein (MBP ScATE1-N; Supplementary Fig. 9a), an effective strategy in previous methods for recombinant production of small proteins/domains that bind [Fe-S] clusters[54]. We then purified this fusion protein (Supplementary Fig. 9b) and subjected it to chemical reconstitution under anoxic

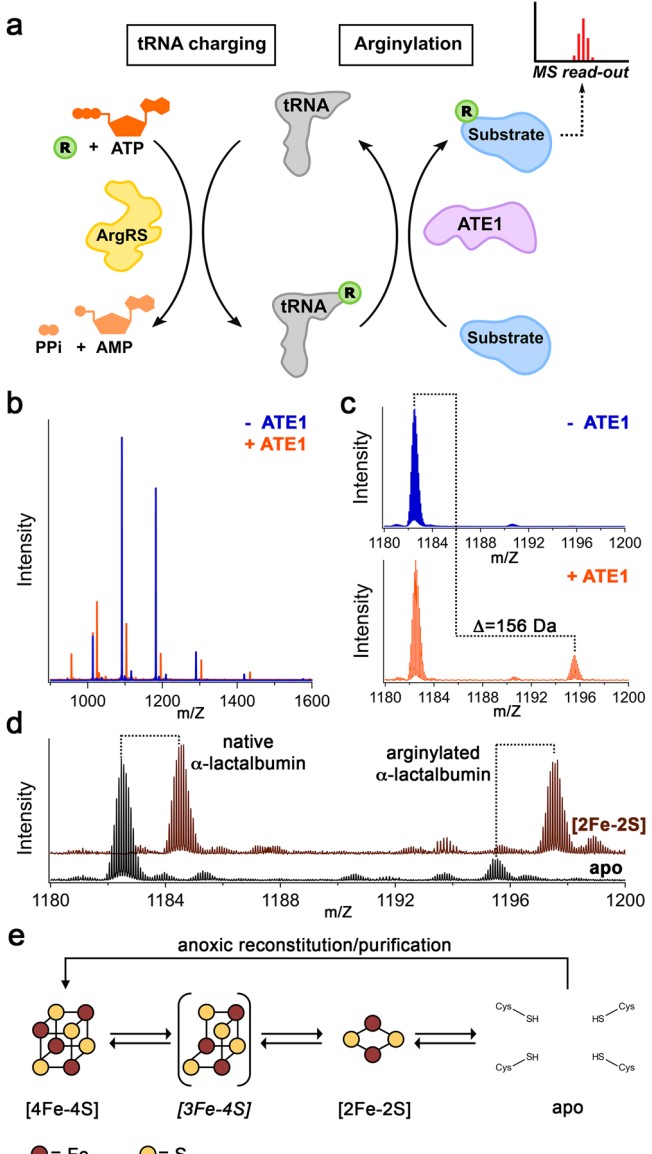

**Fig. 4 | *Sc*ATE1-catalyzed arginylation is affected by the [Fe-S] cluster. a** Cartoon schematic of the recombinant arginylation assay. The arginylated tRNA (Arg tRNA^Arg) is generated by arginine synthetase (ArgRS) aminoacylation. Arginylation of the substrate is catalyzed by ATE1, which is characterized by mass spectrometry. **b** The arginylation of a model substrate (α-lactalbumin) is *Sc*ATE1-dependent, as evidenced by the shift in the top-down mass spectral envelope of α-lactalbumin in the absence of *Sc*ATE1 (blue) and the presence of *Sc*ATE1 (orange). **c** Close-up of the mass spectral window from 1014 m/Z to 1028 m/Z. The change in molecular weight (Δ = 156 g/mol (Da)) is consistent with a single arginylation event after considering the charge (+14) of the polypeptide. **d** In the presence of the [2Fe-2S]^{2+} cluster-bound form of *Sc*ATE1 (brown spectrum), the extent of arginylation of α-lactalbumin is increased *ca.* two-fold on average compared to apo *Sc*ATE1 (black spectrum). For ease of interpretation, the brown spectrum is offset by 2 m/Z relative to the black spectrum. **e** Cartoon synopsis of the cluster-bound forms of *Sc*ATE1. The [4Fe-4S]- and [2Fe-2S]-cluster-bound forms may be interconverted in an O₂-dependent manner, through a [3Fe-4S] intermediate. We believe a [4Fe-4S] cluster is operative under anoxic conditions in the cell until a reaction with O₂ converts the cluster to [2Fe-2S], and we observe such behavior when *Sc*ATE1 is purified and/or reconstituted under anoxic conditions and subsequently exposed to O₂. Source data are provided as a Source Data file.

conditions. After anoxic desalting, the MBP *Sc*ATE1-N retained a similar Fe/polypeptide stoichiometry, and the electronic absorption spectrum was nearly superimposable with that of *Sc*ATE1 reconstituted under anoxic conditions (Supplementary Fig. 9c). These

results demonstrate that the N-terminal domain is the site of cluster binding in *Sc*ATE1.

To confirm the involvement of these specific conserved Cys residues in cluster binding, we next generated variant proteins of full-length *Sc*ATE1 in which Cys-to-Ser substitutions were created via site-directed mutagenesis. We created three variant proteins: one variant in which the C^{20}xxC^{23} motif were changed into S^{20}xxS^{23}; one variant in which the C^{94}C^{95} motif were changed into S^{94}S^{95}; and one variant in which both motifs were altered into S^{20}xxS^{23} and S^{94}S^{95}. After expression and purification in the same manner as WT *Sc*ATE1, each variant protein was tested for iron binding under anoxic conditions and the electronic absorption spectroscopic signatures of the [4Fe-4S]^{2+} cluster. Analysis of the S^{20}xxS^{23} variant showed that loss of the CxxC motif dramatically decreased iron binding (1.00 ± 0.80; *n* = 3) and importantly eliminated the spectral features of the [4Fe-4S]^{2+} cluster (Supplementary Fig. 10). In contrast, however, analysis of the S^{94}S^{95} variant showed that iron was still capable of binding, albeit at a reduced capacity (2.19 ± 0.17; *n* = 3); moreover, the electronic absorption spectral features of the [4Fe-4S]^{2+} were diminished, indicating either a different binding mode in this variant or adventitious iron binding (Supplementary Fig. 10). Analysis of the quadruple variant (S^{20}xxS^{23} and S^{94}S^{95}) demonstrated recapitulation of the spectral behavior of the S^{94}S^{95} variant with only minimal iron binding (1.25 ± 0.07; *n* = 3), implying indeed that some adventitious iron may still bind to *Sc*ATE1 in the absence of the cluster-coordinating ligands (Supplementary Fig. 10), which is possible as there 11 additional Cys residues in *Sc*ATE1 aside from the cluster-binding residues. Nevertheless, these data are consistent with very early in vitro studies on *Sc*ATE1 that demonstrated the importance of these four strongly-conserved N-terminal Cys residues (Cys^{20/23/94/95})[65] as well as our own in vivo data (*vide infra*)[65]. Moreover, these results indicate an essential nature of the CxxC motif in metal binding, while all four Cys residues are required to assemble the cluster in its entirety.

**The *Sc*ATE1 [Fe-S] cluster affects in vitro and in vivo arginylation**
To determine whether the presence of the [Fe-S] cluster affected arginylation efficacy, we first tested the activity of our apo *Sc*ATE1 expressed in the absence of cluster building blocks (i.e., ISC, exogenous ferric citrate, and exogenous *L*-Cys). We used a mass spectrometry-based assay that takes advantage of the natural Glu residue on the N-terminus of α-lactalbumin (ALB)[66], allowing us to do a direct, top-down mass spectrometric comparison between the non-arginylated (molecular mass = 14,176 g/mol) and the arginylated ALB (molecular mass = 14,333 g/mol). *Sc*ATE1 produced in the absence of ISC and without media supplementation (≤0.1 Fe ion per polypeptide) was active towards arginylation in the presence of recombinant *E. coli* arginine synthetase (ArgRS) and both *E. coli* and yeast tRNA^Arg (Fig. 4a, b). This reaction was evidenced by the increase in m/Z ratio of observed ALB after the reaction, which corresponds to a mass difference of 156 g/mol (Fig. 4b): plus one Arg residue (174 g/mol) minus one H₂O (18 g/mol) upon peptide bond formation after calculation of the peptide charge (Z = 14). This N-terminal modification is dependent on the presence of *Sc*ATE1, as no arginylation is observed in the absence of *Sc*ATE1 (Fig. 4b, c). These results indicate that our recombinant *Sc*ATE1 is enzymatically competent in the apo form. We then tested for arginylation of ALB in the presence of *Sc*ATE1 with an [Fe-S] cluster. Because of the rapid oxygen-sensitive nature of the [3Fe-4S]^{+}- and [4Fe-4S]^{2+}-bound forms of *Sc*ATE1, we could not assess the arginylation efficacy and confidently guarantee cluster composition. However, we were able to assess the effects of the oxygen-stable [2Fe-2S]^{2+}-bound *Sc*ATE1 on ALB arginylation. Importantly, we observed a *ca.* twofold increase in the ability of the [2Fe-2S]^{2+}-bound *Sc*ATE1 to arginylate ALB when compared to apo *Sc*ATE1 under identical conditions (Fig. 4d; an average of 39.7% arginylation of [2Fe-2S]^{2+}-bound *Sc*ATE1 compared to an average of 20.6% arginylation of apo *Sc*ATE1).

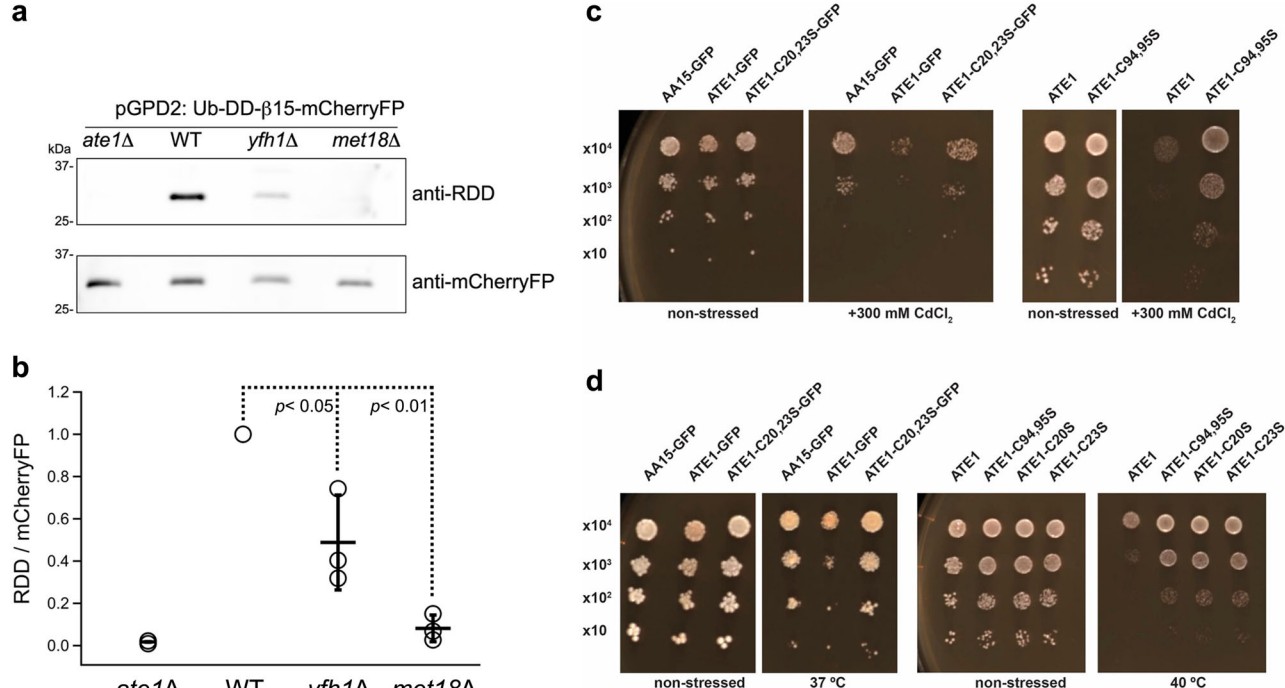

**Fig. 5 | The [Fe-S] cluster biogenesis machinery is linked to in vivo arginylation, and alteration of the ATE1 [Fe-S] cluster-binding capacity affects the yeast stress response.** In panels **a**, **b**, the *yfh1Δ* and *met18Δ S. cerevisiae* carry the arginylation reporter (Ub-DD-β15-mCherryFP), while the wild-type and *ate1Δ* yeasts were also transformed as positive and negative controls. In panel **a**, the signal of arginylated DD-β15 peptide is detected by an antibody specifically recognizing the N-terminal RDD residues. The total level of the reporter protein was measured by an antibody recognizing mCherryFP. Representative images (*n* = 3 independent yeast growths) in panel **a** show the signals detected by anti-RDD and anti-mCherryFP. To avoid signal cross-over, the detections of RDD and mCherryFP were performed on separate membranes with identical loading volumes. Panel **b** shows the ratio between RDD and mCherryFP (*n* = 3 independent yeast growths represented by open circles, the average indicated by a horizontal bar, and error bars representing ±1 standard deviation of the mean). Statistical significance was calculated by the Student's *t*-test (two-sided ANOVA) and $p < 0.01$ was considered highly significant.

Panels **c**, **d** are representative images of yeast containing transiently expressed recombinant ATE1 incubated under non-stressed or stressed conditions. The *ate1Δ* yeasts carry plasmids with a Ura marker containing different versions of ATE1 (or a control). These yeasts were incubated in galactose-containing Ura- SD liquid media to induce protein expression and then inoculated (with serial dilutions) on glucose-containing Ura- SD plates to terminate the expression of ATE1, and yeast were subjected to stressed or non-stressed conditions. For the expression control of GFP-tagged ATE1, AA15-GFP, an arginylation-deficient *ate1Δ* yeast was used. To exclude any impact of the choice of fusion proteins, *Sc*ATE1-6xHis (abbreviated as "ATE1" and identical to that in the bacterial expression system) was also used and was compared to C94,95S, C20S, and C23S variants. As the data indicate, the expression of ATE1-GFP or ATE1-6xHis sensitizes the *ate1Δ* yeast to stress treatments (both panels), while the ATE1 variants with mutations of the putative [Fe-S] binding sites diminish these effects when the yeast are challenged by CdCl2 stress (**c**) or temperature stress (**d**). Source data are provided as a Source Data file.

Next, to demonstrate the connection between the [Fe-S] cluster biogenesis and/or delivery machinery on ATE1 function, we tested the effects of disruptions in the ISC and CIA pathways to in vivo arginylation. Specifically, we employed yeast strains carrying genetic deletions in Yfh1 or Met18, both of which play important functions in [Fe-S] cluster biogenesis. Deletions of these genes are known to minimize (although do not completely abrogate) the corresponding [Fe-S] biogenesis processes[67–69]. Both *yfh1Δ* and *met18Δ S. cerevisiae* strains appear to express ATE1 at levels similar to the wild-type yeast (Supplementary Fig. 11). To measure the in vivo arginylation in these yeast strains, we took advantage of an established reporter assay that utilizes a known arginylation substrate (a peptide derived from β-actin) fused to a fluorescent protein. The plasmid carrying this reporter was used to transform the *yfh1Δ* and *met18Δ* yeast strains, as well as the wild-type and *ate1Δ* yeast strains. The level of in vivo arginylation of the reporter protein was then determined by an antibody specifically recognizing the arginylated form and another antibody against the fluorescent protein, similar as described in past studies from us and others[70–72]. Importantly, we observed that the deletions in *yfh1* or *met18* lead to a significant decrease in in vivo arginylation activity (Fig. 5a, b). Thus, these results demonstrate a link to the cellular [Fe-S] cluster biogenesis machinery and ATE1 function, and these data further support our hypothesis that [Fe-S] clusters are linked to post-translational arginylation, both in vitro and in vivo.

## Alteration of the *Sc*ATE1 [Fe-S] cluster affects the yeast stress response

ATE1 is known to be essential for mediating the stress response for several organisms, including yeast. To test the significance of [Fe-S] cluster binding to ATE1's function in vivo, we generated plasmids with mutated Cys→Ser residues (20, 23, 94, and 95) within *Sc*ATE1 that are essential for [Fe-S] binding, which is expected to diminish the interaction with [Fe-S] with minimal disruption of the protein structure (*vide supra*). We then transformed *ate1Δ* yeast lacking endogenous ATE1 with galactose-inducible yeast expression vectors carrying either the WT, mutant ATE1, or an expression control vector (an irrelevant GFP fusion protein). To avoid the induction of cell death by excessive ATE1 production, the expression of the recombinant proteins was induced for short durations that are known not to lead to significant cell death[70]. The yeast containing the expressed WT ATE1, mutant ATE1, or control (expression verified by Western blotting in ref. [70]), were then challenged with various stress conditions, including thiophilic metals (Fig. 5c) and high temperatures (Fig. 5d), which are known to trigger the yeast stress response in an ATE1-dependent manner[70]. As expected, we found that the expression of WT ATE1 re-sensitizes the *ate1Δ* yeasts to stressing factors (Fig. 5c, d), while the mutation of any of the four Cys residues diminishes the stress response, as expected for a non- or reduced-functional ATE1. This result is similar to the transient expression of

the vector control (Fig. 5c, d), whereas the transient expression of WT ATE1 showed the expected stress response (Fig. 5c, d). Taken together, these findings are consistent with our hypothesis that ATE1 binds an [Fe-S] cluster that regulates function in vivo, and these results show that alteration of the [Fe-S] cluster-binding capacity dramatically affects the normal function of ATE1 in yeast.

## Heme binding induces ATE1 aggregation and inactivation

We next sought to rectify our observations that ScATE1 binds an [Fe-S] cluster with the previous implication that ATE1 could be a heme-binding protein[38]. Despite our efforts using media supplementation, under no circumstances did we observe the incorporation of heme into ScATE1 (*vide supra*). We then attempted in vitro reconstitution by titrating hemin into ScATE1. Over the course of adding 1 mole eq. to 5 mole eq. hemin, we commonly observed visible protein precipitation. Once centrifuged, this protein pellet was colored, indicating an unknown amount of co-precipitation of hemin and ScATE1. For the fraction of protein that did not precipitate, we desalted to remove any excess hemin in the solution. Intriguingly, the absorbance spectrum of the remaining heme-bound ScATE1 (Supplementary Fig. 12) is strikingly similar to the spectra of non-specific heme-protein interactions, as previously determined[73]. Furthermore, we found that the pools of protein with the highest ratio of heme to ScATE1 ($\approx$ 1, estimated by the ratio of $A_{280}/A_{400}$) appear to be aggregated, as determined by size-exclusion chromatography (SEC) (Supplementary Fig. 13a, b). In fact, this aggregation appears to be heme-dependent, and we observe a clear shift to higher-order aggregation based on SEC as we titrated hemin into ScATE1 (Supplementary Fig. 13a, b). Moreover, these pools of aggregated ScATE1 were inactive as determined by our MS assay, and as previously observed for mouse (*Mus musculus*) ATE1-1[38]. Thus, these data indicate that heme binding to ScATE1 may be an off-target effect that is distinct from that of cluster-binding and results in deactivation via protein aggregation. We believe this same heme-induced aggregation may have been operative in previous studies of heme-binding inactivation of MmATE1-1[38], which we believe is also an [Fe-S] protein (*vide infra*).

## The ATE1 [Fe-S] cluster is evolutionarily conserved

Finally, because our partial sequence alignments reveal strong conservation of cluster-binding motifs within the N-terminal region of ATE1s across evolutionary space (Supplementary Fig. 8), we believe these observations strongly indicate [Fe-S] clusters are a common feature of eukaryotic ATE1s. To strengthen this argument, we expressed and purified tagged MmATE1-1, which was previously studied in the context of heme binding[38]. When expressed in the presence of exogenous ferric citrate and *L*-Cys in the growth media and the concentrated, the purified protein exhibited an intense brown color, indicative of [Fe-S] clusters. It is clear from the EA spectrum of MmATE1-1 purified under oxic conditions that a $[2Fe-2S]^{2+}$ cluster nearly identical to that observed in ScATE1 is present (Supplementary Fig. 14), and reconstitution under anoxic conditions results in a spectrum that looks nearly identical to that of the $[4Fe-4S]^{2+}$ cluster bound to ScATE1 and MBP ScATE1-N reconstituted under identical conditions (Supplementary Fig. 14).

We then characterized the cluster-bound MmATE1-1 using XAS and EPR spectroscopies. Like that of ScATE1, XAS characterization of the anoxic reconstituted cluster-bound MmATE1-1 revealed a distinct pre-edge feature at 7112.5 eV and a first inflection point of the edge at $\approx$7120 eV. Simulations of the EXAFS data of anoxically-reconstituted MmATE1-1 are wholly consistent with the presence of a [4Fe-4S] cluster (Supplementary Table 1 and Supplementary Fig. 16). This assignment is confirmed via EPR spectroscopy, which revealed a rhombic signal with $g_1 \approx 2.04$, $g_2 \approx 1.93$, and $g_3 \approx 1.89$, consistent with a $[4Fe-4S]^{+}$ $S = \frac{1}{2}$ species after reduction with sodium dithionite (Supplementary Fig. 16). Fascinatingly, after reaction of this [4Fe-4S] cluster with $O_2$ and

desalting to remove lost iron, we observe the clear presence of $[3Fe-4S]^{+}$ $S = \frac{1}{2}$ species due to the strong axial signal with $g_{\parallel} \approx 2.01$ and $g_{\perp} \approx 1.97$. Finally, EXAFS analyses indicate the presence of a mixture of [3Fe-4S] and [2Fe-2S] in MmATE1-1 data of reconstituted and $O_2$-reacted MmATE1-1 (Supplementary Table 1; Supplementary Fig. 15). While there are some modest differences in sequence among the four major MmATE1 isoforms (Supplementary Figs. 1, 8), we do not believe these will lead to major differences in cluster behavior. In particular, the difference in the 1A *vs.* 1B exon inclusion leads to only a minor difference in the length (seven amino acids) of the extreme N-terminus of the protein (Supplementary Fig. 8). While there is some difference in the flanking sequence of the CxxC motif, the CxxC and CC motifs still align strongly across all eukaryotes, and the intervening length of the polypeptide between these regions only differs by a few amino acids in length (Supplementary Fig. 8). Thus, this extensive in vitro characterization of the cluster-bound MmATE1-1 and its mimicry of the behavior ScATE1 demonstrate that [Fe-S] clusters are a general feature eukaryotic ATE1s, and that reaction of the bound [4Fe-4S] cluster proceeds through a [3Fe-4S] cluster intermediate before conversion to a [2Fe-2S] cluster (Fig. 4), commonly observed for other $O_2$-sensing [Fe-S] cluster proteins[54,74].

## Discussion

ATE1s are critical eukaryotic enzymes that arginylate a diverse set of protein targets essential for general cellular homeostasis and the function of higher-order eukaryotes[4]. The process of arginylation may regulate protein function in one of two major ways. In a non-degradative manner, arginylation may stabilize proteins or even change their activity and/or structure. For example, α-synuclein and β-amyloid, both of which are proteins implicated in neurological diseases, are stabilized and have lower aggregation propensities when arginylated[28,75]. In another example, arginylation can change oligomerization of essential structural proteins, such as the cytoskeletal protein β-actin[27,76]. In contrast, arginylation can also target proteins for degradation via the N-degron pathway. Once arginylated, E3 ubiquitin ligases of the ubiquitin-proteasome system (UPS) ubiquitinate target polypeptides for cellular clearance. While the exact extent of the true intracellular N-degron arginylome remains to be uncovered, proteins linked to the Arg N-degron pathway are implicated in the yeast stress response[70], plant growth[31], and even mammalian cardiovascular[9,19] and neurological[28] development. Thus, the fidelity of post-translational arginylation, which may be achieved through multiple levels and types of regulatory processes, is paramount.

Previous results have indicated that ATE1-connected pathways may be subject to different methods of regulation, such as small diatomic molecules, cofactors, and even protein–protein interactions. In plants, it is now well-established that the response to hypoxia/$O_2$ is connected to the N-degron pathway[39,77,78]. Group VII ethylene response factors (ERFs)—proteins that regulate the plant hypoxic response—were shown to be targets of the N-degron pathway in an $O_2$-dependent manner. Work has demonstrated that several of the group VII ERFs have N-terminal Met-Cys sequences, and ERF protein degradation is linked to the oxidation of the N-terminal Cys residue[79] (presumably after aminopeptidase cleavage) dependent on oxygen availability and the enzymatic capabilities of plant CDOs[24]. These findings connect the concentration of intracellular $O_2$ and the N-degron pathway in plants. A similar connection has been made between NO availability and the N-degron pathway in mammals, where NO has been shown to oxidize N-terminal Cys residues directly rather than enzymatically, rendering secondary destabilizing residues that are recognized by ATE1 for arginylation[18]. NO-linked oxidation of N-terminal Cys residues has been shown to have dramatic effects on the degradation of regulators of G-protein signaling (RGSs), which are critical for cardiovascular maturation in mammals[18]. Although these are indirect regulatory responses (occurring at the protein substrate that is the target of

arginylation), one study has shown that *Mm*ATE1-1 may be inactivated directly by the presence of ferric heme through the formation of a strained disulfide, which presumably alters protein structure[38]. Furthermore, ATE1 may be upregulated through direct protein–protein interactions with a partner protein present in higher-order eukaryotes known as Liat1[80]. The molecular details of this protein–protein interaction remain to be fully described; however, it was demonstrated that binding of this small protein to certain ATE1 isoforms increased the arginylation of a model ATE1 substrate in vitro[80]. Thus, given the multi-layered control of the Arg N-degron pathway, it is perhaps unsurprising that ATE1 itself may also bind and utilize a regulatory cofactor.

In this work, we have discovered that ATE1s bind [Fe-S] clusters, expanding the regulatory complexity of arginylation. Our initial focus on the heme-binding properties of ATE1 revealed instead that ATE1 binds an [Fe-S] cluster. Purified under oxic conditions, our data revealed the presence of a $[2Fe-2S]^{2+}$ cluster (Fig. 4e), which has increased activity towards the in vitro N-terminal arginylation of a model ATE1 substrate. To determine if a higher-order cluster could form, anoxic reconstitution of ATE1 yielded a characteristic $[4Fe-4S]^{2+}$ cluster (Fig. 4e), and these results were recapitulated by anoxic purification of ATE1, indicating that a $[4Fe-4S]^{2+}$ cluster could be formed in vivo. Furthermore, the *Sc*ATE1 cluster is oxygen-sensitive, displaying rapid decomposition from $[4Fe-4S]^{2+}$ to $[2Fe-2S]^{2+}$ through a $[3Fe-4S]^{+}$ intermediate (Fig. 4e) upon exposure to $O_2$. We subsequently mapped the location of the oxygen-sensitive ATE1 [Fe-S] cluster to the N-terminal domain, which is evolutionarily conserved and contains four key Cys residues that are both important for arginylation activity and cluster binding. Importantly, to confirm the physiological relevance of the cluster, we demonstrated that disruption in the [Fe-S] cluster biogenesis and delivery machineries significantly alters in vivo arginylation. Additionally, we tested the effects of mutation of these four key Cys residues, and using transient expression in yeast, we found that loss of [Fe-S] cluster-binding attenuates the stress response in *S. cerevisiae*. Finally, consistent with our bioinformatic data showing the conservation of these key Cys residues across eukaryotes, we demonstrated that *Mm*ATE1-1 is also capable of binding an $O_2$-sensitive [Fe-S] cluster, and that the cluster has spectroscopic parameters nearly identical to the cluster bound to the yeast protein. Given these current and previous results, we propose that an ATE1 [Fe-S] cluster is present in all eukaryotes, including humans.

These findings shed light onto early studies that demonstrated the importance of these Cys residues to ATE1 function[34,65,81]. For example, these precise residues have been shown to be important for the maximal activity of *S. cerevisiae* ATE1, and deletions or alterations of these key Cys residues result in a major decrease in ATE1 activity[34,65,81]. These observations led to the hypothesis that one or several of these Cys residues were part of the enzymatic active site[65,81]. Our study clearly demonstrates instead that this loss in activity is due to the loss of the [Fe-S] cluster-binding function of ATE1 (likely an allosteric site), not a loss of the ATE1 active site per se.

It is tempting to speculate that the presence of the ATE1 [Fe-S] cluster and its $O_2$-sensitivity may be linked to function, especially given the capabilities of the N-degron pathway to sense NO and $O_2$ (*vide supra*). The binding of an [Fe-S] cluster by ATE1 would be well matched to these known functions of the N-degron pathway, as [Fe-S] proteins are either known or speculated to sense $O_2$, NO, and even the redox state of the cell[54,55,82–84]. We do not have evidence that this cluster functions as an electron transport site and there is no need for electron transport in the arginylation reaction (Fig. 4a); furthermore, the reduction of the cluster in ATE1 is relatively slow and results in instability and loss of cluster loading. Instead, we posit that the evolutionarily conserved [Fe-S] cluster within the ATE1 N-terminal domain may function to sense $O_2$, NO, and even respond to oxidative stress. Under such conditions, N-terminal Cys residues may be oxidized, and/

or proteins may be misfolded, resulting in a greater need to flag and degrade dysfunctional polypeptides. We imagine that altered compositions of the cluster may transmit this information through the N-terminal domain to the catalytic C-terminal domain, potentially changing the rate of arginylation and even arginylation specificity. This function, which warrants future in-depth studies, would be entirely consistent with previous results demonstrating that the Arg N-degron pathway is capable of sensing and responding to changes in intracellular $O_2$ and NO levels.

We believe that our proposed [Fe-S] cluster-based sensor hypothesis is distinct from previous observations of hemin interactions with ATE1. The prior studies of *Mm*ATE1-1 mapped the ATE1-heme interaction to residue $Cys^{411}$ [38], located within the C-terminal domain of ATE1 and distinct from the Cys residues involved in [Fe-S] coordination. However, it is not uncommon for exposed Cys residues to interact with heme in an adventitious manner[73], and *Mm*ATE1-1 appeared to harbor multiple potential heme-interacting sites with very modest binding affinities[38]. The stated causal relationship of decreased arginylation of *Mm*ATE1-1 was hemin-derived oxidation of two adjacent Cys residues to an unusually strained disulfide in the N-terminal domain, but it is difficult to rectify how this occurs via heme binding in the C-terminal catalytic region. It is commonly accepted that there is no free heme within the cell under homeostatic conditions (*i.e.*, heme is either bound tightly by hemoproteins, heme chaperones, or heme scavengers)[85], but it is possible that heme may be spuriously released under dyshomeostasis during oxidative stress or infection within the cell, a situation that could lead to the observed decrease in ATE1 activity. Contrary to previous work[38], we find that ATE1 appears to bind heme non-specifically and induces protein aggregation leading to inactivation, suggesting the heme-based inactivation may be irrelevant to ATE1's native function. Based on our results, we present the compelling discovery that [Fe-S] clusters are the operative cofactors that bind and give function to the intriguing ATE1 N-terminal domain. This discovery opens many additional avenues of exploration within this family of essential eukaryotic enzymes.

## Methods
### Materials
Certain commercial entities, equipment or materials may be identified in this document to describe an experimental procedure or concept adequately. Such identification is not intended to imply recommendation or endorsement by the National Institute of Standards and Technology, nor is it intended to imply that the entities, materials or equipment are necessarily the best available for the purpose. Modified forms of the BL21(DE3) and C43(DE3) *E. coli* expression cell lines in which the gene for the multidrug exporter AcrB (a common *E. coli* contaminant) had been deleted (BL21(DE3) Δ*acrB* and C43(DE3) Δ*acrB*, respectively) were generous gifts of Prof. Edward Yu (Case Western Reserve University). All materials used for buffer preparation, protein expression, and protein purification were purchased from commercial vendors and were used as received. All unique biological materials (e.g., expression plasmids) are readily available from the authors upon request.

### Cloning, expression, and oxic purification of *Sc*ATE1 constructs
DNA encoding for the gene corresponding to the single-isoform arginine transferase (ATE1) from *S. cerevisiae* (Uniprot identifier P16639) (*Sc*ATE1) was commercially synthesized with an additionally engineered DNA sequence encoding a C-terminal TEV-protease cleavage site (ENLYFQS). This gene was subcloned into the pET-21a(+) expression plasmid using the NdeI and XhoI restriction sites, encoding a C-terminal $(His)_6$ affinity tag when read in-frame. The complete expression plasmid was transformed into chemically competent BL21(DE3) Δ*acrB* cells, spread onto Luria-Bertani (LB) agar plates supplemented with 100 µg/mL ampicillin, and grown overnight at

37 °C. Colonies from these plates served as the source of *E. coli* for small-scale starter cultures (generally 100 mL LB supplemented with 100 µg/mL ampicillin). Large-scale expression of *Sc*ATE1 was accomplished in 12 baffled flasks, each containing 1 L sterile LB supplemented with 100 µg/mL (final) ampicillin and inoculated with a pre-culture. Cells were grown by incubating these flasks at 37 °C with shaking until the optical density at 600 nm ($OD_{600}$) reached ≈0.6 to 0.8. The flasks containing cells and media were then chilled to 4 °C for 2 h, after which protein expression was induced by the addition of isopropyl β-D-l-thiogalactopyranoside (IPTG) to a final concentration of 1 mmol/L. The temperature of the incubator shaker was lowered to 18 °C with continued shaking. After *ca.* 18 h to 20 h, cells were harvested by centrifugation at 4800 × *g*, 10 min, 4 °C. Cell pellets were subsequently resuspended in resuspension buffer (50 mmol/L Tris, pH 7.5, 100 mmol/L NaCl, 0.7 mol/L glycerol), flash-frozen on $N_{2(l)}$, and stored at −80 °C until further use.

A similar approach was taken to generate constructs of the *Sc*ATE1-N terminal domain (*Sc*ATE1-N) and a maltose-binding protein fusion of the *Sc*ATE1-N terminal domain (MBP *Sc*ATE1-N). DNA encoding for the genes of the *Sc*ATE1-N-terminal domain (residues 1–146), with additionally engineered DNA sequences encoding for a C-terminal TEV-protease cleavage site (ENLYFQG) or with an additionally engineered DNA sequence encoding for an N-terminal maltose-binding protein sequence (based on Uniprot ID P0AEX9: *Escherichia coli* (K-12) *malE* gene product) followed by a Tobacco Etch Virus (TEV)-protease cleavage site, was synthesized. For the former approach, the gene was subcloned into the pET-21a(+) expression plasmid using the NdeI and XhoI restriction sites, encoding for a C-terminal (His)$_6$ affinity tag when read in-frame. For the latter approach, the gene was subcloned into the pET-45b(+) expression plasmid using the PmlI and PacI restriction sites, encoding for an N-terminal (His)$_6$ affinity tag followed by maltose-binding protein when read in-frame. Transformations, cell growths, and cell resuspensions for both constructs were identical to those of the WT construct.

All steps for the purification of *Sc*ATE1 were performed at 4 °C unless otherwise noted. Briefly, frozen cells were thawed and stirred until the solution was homogeneous. Solid phenylmethylsulfonyl fluoride (PMSF; 50 to 100 mg), and a solution of tris(2-carboxyethyl) phosphine hydrochloride (TCEP; 1 mmol/L final concentration) were added immediately prior to cellular disruption using an ultrasonic cell disruptor set to a maximal amplitude of 70%, 30 s pulse on, 30 s pulse off, for a total pulse on time of 12 min. Cellular debris was cleared by ultracentrifugation at 163,000 × *g* for 1 h. The supernatant was then applied to a 5 mL metal affinity column that had been charged with Ni$^{2+}$ and equilibrated with 8 column volumes of wash buffer (50 mmol/L Tris, pH 8.0, 300 mmol/L NaCl, 1 mmol/L TCEP, and 1.4 mol/L glycerol) with an additional 21 mmol/L imidazole. The column was then washed with 12 column volumes (CVs) of wash buffer with an additional 30 mmol/L imidazole. Protein was then eluted with a wash buffer containing an additional 300 mmol/L imidazole. Fractions were concentrated using a 15 mL 30 kg/mol (30 kDa) molecular-weight cutoff (MWCO) spin concentrator. Protein was then applied to a 120 mL gel filtration column that had been pre-equilibrated with 50 mmol/L Tris, pH 7.5, 100 mmol/L KCl, 1 mmol/L dithiothreitol (DTT), and 0.7 mol/L glycerol. The eluted fractions of the protein corresponding to the protein monomer (≈60,000 g/mol; ≈60 kDa), were pooled and concentrated with a 4 mL 30 kg/mol (30 kDa) MWCO spin concentrator. Protein concentration was determined using the Lowry assay, and purity was assessed via SDS-PAGE analysis (acrylamide mass fraction of 15%). To generate the variant proteins, site-directed mutagenesis was employed utilizing the Agilent QuikChange Site-Directed Mutagenesis kit (a list of primers used in this study is provided in Supplementary Table 2).

To purify MBP *Sc*ATE1-N, a similar protocol was followed for the WT construct with the following modifications, similarly described in ref. [54]. Briefly, the supernatant was applied to two tandem 5 mL amylose columns that had been pre-equilibrated with five CVs of wash buffer (25 mmol/L Tris, pH 7.5, 200 mmol/L NaCl, 0.7 mol/L glycerol, 1 mmol/L TCEP). The column was then washed with 20 CVs of wash buffer. Protein was then eluted by a wash buffer containing 10 mmol/L maltose. Fractions were concentrated using a 15 mL 30 kg/mol (30 kDa) MWCO spin concentrator. Protein concentration was determined using the Lowry assay, and purity was assessed via SDS-PAGE (acrylamide mass fraction of 15%) analysis.

To express *Sc*ATE1 in the presence of heme and/or iron-sulfur precursors, the protein was expressed as previously described (*vide supra*) with some modifications. To encourage the incorporation of heme, immediately prior to the addition of IPTG, δ-aminolevulinic acid (1 mmol/L final concentration) and ferric citrate (125 µmol/L final concentration) were added to the expression cultures. To encourage the presence of iron-sulfur clusters, immediately prior to the addition of IPTG, L-Cys (167 mg/L final concentration) and ferric citrate (125 µmol/L final concentration) were added to the expression cultures. All subsequent steps were performed as previously described (*vide supra*).

To express *Sc*ATE1 in the presence of the *E. coli* iron-sulfur cluster (ISC) biosynthesis machinery, the *Sc*ATE1 expression plasmid (encoding for ampicillin resistance) was transformed into chemically competent C43 (DE3) Δ*acrB* cells bearing the pACYC184*iscS-fdx* plasmid (encoding for chloramphenicol resistance and constitutively expressed), spread onto LB agar plates supplemented with 100 µg/mL ampicillin and 34 µg/mL chloramphenicol, and grown overnight at 37 °C. Large-scale expression was accomplished as above with the following modifications: immediately prior to the addition of IPTG, L-Cys (167 mg/L final concentration) and ferric citrate (125 µmol/L final concentration) were added to the expression cultures. After *ca.* 18 to 20 h, cells were harvested, lysed, and protein was purified as previously described. There was no dramatic change in protein yield, purity, or homogeneity; however, the purified protein was deeply colored even after size-exclusion chromatography.

Western blotting was used to confirm the presence of the (His)$_6$-tagged *Sc*ATE1 constructs. Briefly, protein samples were run on SDS-PAGE (acrylamide mass fraction of 15%) and then transferred to a PVDF membrane overnight. Blocking of the membrane was done at room temperature by the addition of instant milk solution (milk mass fraction of 5%) in phosphate-buffered saline with Tween 20 (PBST; Tween 20 volume fraction of 1%) with rocking for 1 h. The membrane was then washed with PBST, and an anti-(His)$_6$ antibody (Millipore-Sigma, catalog # A7058-1VL) that was diluted *ca.* 1:6000 (volume fraction) into an instant milk solution (milk mass fraction of 1%) in PBST and incubated with rocking for 1 h. The membrane was then washed with PBST and developed using a chromogenic detection of horseradish peroxidase kit.

## Anoxic expression and purification of *Sc*ATE1

Overnight starting cultures of *E. coli* BL21(DE3) Δ*iscR* (an initial gift from Prof. Patrik R. Jones, Imperial College) containing the pET21(+) *Sc*ATE1 plasmid were used to inoculate Terrific Broth (TB) medium. TB medium was supplemented with kanamycin (50 µg/mL), ampicillin (100 µg/mL) and ferric ammonium citrate (2 mmol/L). Cells were cultivated under oxic conditions at 37 °C with shaking until $OD_{600}$ reached ≈2. For anoxic cells, growth cultures were then moved to an anoxic glove box containing an $H_2/N_2$ atmosphere and operating at <7 mg/m$^3$ (5 ppm) $O_2$. Gene expression was induced by adding IPTG to a final concentration of 0.5 mmol/L. To facilitate [Fe-S] cluster assembly and anaerobic metabolism, 2 mmol/L L-cysteine as well as 25 mmol/L sodium fumarate were added. Cultures were stirred on a magnetic stirrer at RT for 20 h following induction.

Cells were harvested by spinning for 10 min at 6000 × $g$ and 4 °C, while cultures were covered with Ar$_{(g)}$ to maintain anaerobic conditions. For cell lysis, cells were resuspended in 5 mL buffer A (50 mmol/L Tris, pH 8.0, 300 mmol/L NaCl, 1.4 mol/L glycerol) per gram of cell pellet. Protease inhibitor tablets were added as needed. After stirring at RT for 20 min under anaerobic conditions the Ar$_{(g)}$ covered suspension was sonicated for 20 min with an amplitude of 60% and a pulse of 1 s every 3 s using an ultrasonic cell disrupter. Argon-covered lysates were clarified by centrifugation at 40,000 × $g$.

$Sc$ATE1 purification was carried out under anoxic conditions at RT. The anoxic lysate was applied to a metal affinity column with a bed volume of 5 mL equilibrated with buffer A. The column was then washed with ten CVs buffer A and ten CVs of a mixture of buffers A and B (buffer B volume fraction of 10%, 50 mmol/L Tris, pH 8.0, 300 mmol/L NaCl, 1.4 mol/L glycerol, 300 mmol/L imidazole) before the target protein was eluted with only buffer B. Fractions containing the brown-colored target protein were pooled and subsequently applied to a 24 mL size-exclusion column pre-equilibrated in 25 mmol/L Tris, pH 7.5, 0.7 mol/L, 100 mmol/L KCl. Eluted monomeric protein was then further used for electronic absorption and electron paramagnetic spectroscopies (*vide infra*).

## Purification of *Mm*ATE1-1

The kanamycin-resistant pET29a(+) expression plasmid encoding for N-terminally tagged mouse (*Mus musculus*) ATE1-1 (Uniprot ID Q9Z2A5) was prepared as previously described[33]. The expression plasmid was transformed into chemically competent BL21 Codon Plus cells (Agilent part # 230245), spread onto LB agar plates supplemented with 100 μg/mL kanamycin and 50 μg/mL chloramphenicol and grown overnight at 37 °C. Expression of *Mm*ATE1-1 was identical to that of *Sc*ATE1 (*vide supra*) and cells were grown in the presence of supplemented *L*-Cys and ferric citrate, as described (*vide supra*), except that the antibiotics kanamycin and chloramphenicol were supplemented to 100 and 50 μg/mL (final concentrations), respectively, instead of ampicillin. The purification of *Mm*ATE1-1 was identical to that of *Sc*ATE1 (*vide supra*).

## Cloning, expression, and purification of *Ec*ArgRS

DNA encoding for the gene corresponding to the Arg tRNA synthetase from *E. coli* (strain K12; Uniprot identifier P11875) (*Ec*ArgRS) was commercially synthesized with an additionally engineered DNA sequence encoding for a C-terminal TEV-protease cleavage site (ENLYFQS). This gene was subcloned into the pET-21a(+) expression plasmid using the NdeI and XhoI restriction sites, encoding for a C-terminal (His)$_6$ affinity tag when read in-frame. The complete expression plasmid was transformed into chemically competent BL21(DE3) Δ*acrB* cells, spread onto Luria-Bertani (LB) agar plates supplemented with 100 μg/mL ampicillin, and grown overnight at 37 °C. Colonies from these plates served as the source of *E. coli* for small-scale starter cultures. *Ec*ArgRS was purified in essentially the same manner as *Sc*ATE1 with the following modification: during size-exclusion chromatography, a buffer composed of 25 mmol/L Tris, pH 7.5, 100 mmol/L NaCl, 1 mmol/L TCEP, 0.35 mol/L glycerol was used. Under these conditions, *Ec*ArgRS eluted on a 120 mL gel filtration column as two species: one aggregated species and one apparent hexameric species. Only the hexameric species was pooled for downstream arginylation assays. Protein concentration was determined using the Lowry assay, and purity was assessed via SDS-PAGE analysis (acrylamide mass fraction of 15%).

## Yeast strains

All *S. cerevisiae* strains used in this study were obtained from Open Biosystems (now part of PerkinElmer). The *ate1*Δ strain carries a null *ate1*Δ:KanMX cassette in the parental wild-types BY4741 (*MATa his3Δ1 leu2Δ0 met15Δ0 ura3Δ0*) (catalog ID YSC6273-201936070). This strain was validated in ref. [70]. The *met18*Δ and *yfh1*Δ strains (catalog IDs

YSC6273-201925518 and YSC6274-201926392, respectively) are in BY4741 and BY4743 (*MATa/α his3Δ1/his3Δ1 leu2Δ0/leu2Δ0 LYS2/lys2Δ0 met15Δ0/MET15 ura3Δ0/ura3Δ0*) background, respectively. These strains were validated in past studies[67–69].

## Preparation of plasmids for yeast expression

The preparation of most of the plasmids were performed as described in ref. [70]. The *Sc*ATE1 gene (wild-type or mutant) were cloned in pYES2 vector (Life Technologies, # V825-20) at the Kpn1 and Xba1 sites, which is a galactose-inducible yeast expression vector containing a GAL1 promoter. A list of primers for cloning *Sc*ATE1 with a 6xHis tag is provided in Supplementary Table 2. Based on this approach, the in vivo expressed yeast *Sc*ATE1 is C-terminally fused with a TEV-cleavage site and a 6xHis tag, exactly as the recombinant protein expressed in the bacterial system. The pYES-Ub-AA15-GFP construct was cloned similarly as described in ref. [70], except that a span of 15 amino acids derived from the residue 1903 of mouse talin-1 (in the sequence of AAVVAAE-NEEIGAHIK) was used as a low-sensitivity reporter for arginylation activity. This reporter is not expected to be activated in the arginylation-deficient *ate1*Δ yeasts and therefore is used as an irrelevant expression control for GFP-fused proteins. The reporter protein for in vivo arginylation, DD-β15-mCherryFP, was cloned similarly as described in our past work[70]. In brief, the coding region of 15 residues of β-actin (DDIAALVVDNGSGMC) was fused with an N-terminal ubiquitin and a C-terminal mCherryFP. This region was then cloned into a yeast expression vector pGPD2 (Addgene # 43972) using the EcoR1 and Xho1 restriction sites. During expression in the yeast, the N-terminal ubiquitin is spontaneously removed by the endogenous de-ubiquitination machinery, leaving the β-actin peptide exposed for in vivo arginylation (detected as described below).

## Yeast culture

Yeast culture media were prepared as follows: SD (Synthetic Defined) dropout base medium without uracil (Ura⁻) or sugar: yeast nitrogen base (1.7 g), ammonium sulfate (5 g), and required amino acids (Sunrise Science Products Catalog #: 1004-010; 50 mg). The medium was supplemented with 2% dextrose/raffinose/galactose as necessary. For solid media plates, 2% agar was dissolved in the liquid media. For non-stress conditions, yeast strains were incubated at 30 °C unless otherwise indicated.

## Transient expression of ATE1 in *ate1*Δ yeast cells and stress treatment

The yeast strains were grown in YPD media and then transformed with the lithium-based method as in ref. [70]. The yeast carrying the Ura selection marker were then inoculated in Ura⁻ SD plates to allow successful transformants to grow into single colonies for selection. To transiently express recombinant ATE1 (WT or variant) or the control protein Ub-AA15-GFP, the yeast carrying the galactose-inducible vectors were first grown in dextrose-containing Ura⁻ SD media. After the yeast grew to the log phase (OD$_{600}$ ≈ 0.2–1.0), they were washed with sterile water three times and then transferred to raffinose-containing Ura⁻ SD media for at least 6 h. The yeast were then washed with sterile water three times again and then inoculated in galactose-containing Ura⁻ SD media to induce protein expression for 8–10 h, which is not expected to lead to a significant amount of cell death as shown in ref. [70].

The yeast with induced expression were then washed with sterile water. The culture densities were first measured by OD$_{600}$. As a validation, direct counting of the live cells was then performed with an automated cell counter TC20 (Bio-Rad) with the aid of cell viability dye Trypan Blue. The culture densities of various samples were adjusted to be equal for subsequential serial dilution and plotting on glucose-containing Ura⁻ SD plates for stress treatments such as CdCl$_2$ or high temperature (37 or 40 °C) as described in ref. [70]. After

2–3 days, the images of the yeast colonies (in stressed or non-stressed conditions) were taken at the same timepoint using a GE Amersham Imager 600.

## In vivo arginylation assays in yeast

The plasmid carrying the coding sequence of the reporter protein, DD-β15-mCherryFP, was used to transform yeast strains. Positive colonies selected from a Ura⁻ plate were allowed to grow at 30 °C in a shaker until they reached the log phase (OD$_{600}$ ≈ 0.2–1.0). The cells were then harvested and lysed for Western blotting. The signal of the N-terminally arginylated reporter protein was detected by an antibody specifically recognizing the RDD sequence, which was custom ordered by Genscript and validated in our past study[70]. The total level of the reporter protein was probed with an antibody against mCherryFP (Clone 16D7, Catalog# M11217, from Thermo Scientific). The ratio of RDD to mCherryFP was calculated for the level of arginylation, similar as described in our past study[70].

## Iron and sulfide content determination

Iron content was determined spectrophotometrically using a modified version of the ferrozine assay[86,87]. Briefly, the protein was precipitated using 5 mol/L trichloroacetic acid (TCA). The supernatant was decanted and subsequently neutralized with saturated ammonium acetate. To this solution, excess ascorbic acid and 0.30 mmol/L ferrozine (final concentration) were added. Absorbance measurements of samples made in at least triplicate were taken at 562 nm. The concentration of $Fe^{2+}$ was then determined assuming a $Fe^{2+}$-ferrozine complex with an extinction coefficient ($\varepsilon_{562}$) of ≈28 mmol L$^{-1}$ cm$^{-1}$[87] (26.98 ± 0.96 mmol L$^{-1}$ cm$^{-1}$)[86]. The concentration of acid-labile sulfide ($S^{2-}$) was determined based on a modified method of Beinert[49,50]. Briefly, the cluster-containing protein was incubated with 1% (w/v) zinc acetate and 7% (w/v) sodium hydroxide and incubated at room temperature for approximately 15 min. After incubation, 0.1% (w/v) N,N-dimethyl-p-phenylenediamine (dissolved in 5 M HCl) and 10 mM FeCl$_3$ (stabilized in 1 M HCl) were added, mixed, and the solution was incubated at room temperature for approximately 20 min. The chemically-generated methylene blue was measured at 670 nm with an extinction coefficient ($\varepsilon_{670}$) of ≈35 mmol L$^{-1}$ cm$^{-1}$, and the concentration of sulfide in the sample was determined. These data were corrected against residual iron and sulfide present in buffer constituents and using a protein correction factor based on amino acid quantitation (determined by the University of California, Davis proteomics core facility).

## Anoxic reconstitution

Apoprotein samples were reconstituted in an anoxic chamber containing an N$_2$/H$_2$ atmosphere and operating at <7 mg/m$^3$ (5 ppm) O$_2$. Briefly, the protein was brought into the anoxic chamber and allowed to equilibrate with the anaerobic chamber's atmosphere overnight at 6 °C with shaking. Protein was then diluted to 100 μmol/L in reconstitution buffer comprising 25 mmol/L 3-morpholinopropane-1-sulfonic acid (MOPS) or 25 mmol/L Tris, pH 7.5, 300 mmol/L KCl, 1 mmol/L DTT, 0.7 mol/L glycerol. About 10 mmol/L stock FeCl$_3$ was first titrated into the apoprotein until 4 mole eq. to 10 mole eq. had been added (as warranted) with 15 min shaking at 6 °C between the addition of each mole eq. of $Fe^{3+}$. About 10 mmol/L stock Na$_2$S was then titrated into the iron-bound protein in the same manner. Afterward, the protein was allowed to equilibrate with FeCl$_3$ and Na$_2$S overnight at 6 °C with shaking. Particulate matter was removed by first centrifuging at 10,000 × g anoxically for 10 min, 4 °C, and then by filtration through a filter with a pore size of 0.22 μm. Excess iron and sulfide were removed by buffer exchanging three times into fresh buffer. Iron and protein contents were determined as described (vide supra).

## Electronic absorption (EA) and circular dichroism (CD) spectroscopies

Electronic absorption (EA) spectra were recorded at room temperature on a UV-visible spectrophotometer. Samples were contained within a 1 cm UV-transparent cuvette, and data were acquired from 900 to 250 nm with the instrument set to a spectral bandwidth of 2 nm. Heme titrations were carried out similarly to previously-described methods[38]. Circular dichroism (CD) spectra were recorded on a nitrogen-flushed spectropolarimeter operating at room temperature. Samples were contained within a quartz cuvette with a path length of 0.3 cm, and data were acquired from 800 to 300 nm with the instrument set to a spectral bandwidth of 1 nm. The data presented are an average of five scans.

## X-ray absorption spectroscopy (XAS)

Samples containing ≈0.5 to 2 mmol/L iron (final concentration) in buffer plus 3.6 mol/L ethylene glycol were aliquoted either under oxic conditions or anoxically (as warranted) into cells wrapped with tape, flash-frozen in N$_{2(l)}$ and stored at −80 °C until data collection. X-ray absorption spectra (XAS) were collected on beamlines 7-3 and 9-3 at the Stanford Synchrotron Radiation Lightsource (Menlo Park, CA) as replicates when possible. Extended X-ray absorption fine structure (EXAFS) of Fe (7210 eV) was measured using a Si 220 monochromator with crystal orientation φ = 90°. Samples were measured as frozen aqueous glasses at 15 K, and the X-ray absorbance was detected as Kα fluorescence using either a 100-element (beamline 9-3) or 30-element (beamline 7-3) Ge array detector. A Z-1 metal oxide filter (Mn) and Soller slit assembly were placed in front of the detector to attenuate the elastic scatter peak. A sample-appropriate number of scans of a buffer blank were measured at the absorption edge and subtracted from the raw data to produce a flat pre-edge and eliminate residual Mn Kβ fluorescence of the metal oxide filter. Energy calibration was achieved by placing a Fe metal foil between the second and third ionization chambers. Data reduction and background subtraction were performed using EXAFSPAK (Microsoft Windows version)[88]. The data from each detector channel were inspected for dropouts and glitches before being included in the final average. EXAFS simulation was carried out using the program EXCURVE (version 9.2) as previously described[89–91]. The quality of the fits was determined using the least-squares fitting parameter, F, which is defined as:

$$F^2 = \frac{1}{N}\sum_{i=0}^{N} k^6 \left(\chi_i^{theory} - \chi_i^{exp}\right)^2 \qquad (1)$$

and is referred to as the fit index (FI). Low yields precluded the analysis of ScATE1 purified under anoxic conditions.

## Electron Paramagnetic Resonance (EPR) Spectroscopy

Samples containing ≈100 to 600 μmol/L iron (final concentration) in buffer plus 3.6 mol/L ethylene glycol were aliquoted either aerobically or anaerobically (as warranted) into standard quartz X-band electron paramagnetic resonance (EPR) tubes with an outer diameter of 4 mm and flash-frozen in N$_{2(l)}$. Spectra were collected at temperatures indicated in the figure legend using a commercial EPR spectrometer system equipped with a high-sensitivity, TE-mode, CW resonator, and commercial temperature-control unit. The uncertainty on the reported g values is 0.0005, using the manufacturer-reported field (0.08 mT) and frequency (0.00005 GHz) accuracies. The maximum, minimum, and baseline-crossing points of peaks were used to determine magnetic field positions for g values. Calculated g values (from magnetic field values) agree with g values directly reported by the spectral analysis software provided with the commercial instrument to within 0.001.

## NMR spectroscopy

Anoxically reconstituted samples containing *ca*. 200 μmol/L *Sc*ATE1 were exchanged into 50 mmol/L Tris, pD 7.5, 100 mmol/L KCl, 1 mmol/L DTT using a desalting column and loaded into an NMR tube with an outer diameter of 3 mm that was capped to avoid exposure to air. Paramagnetic 1D $^1$H NMR spectra were acquired at 10 °C on a 500 MHz spectrometer equipped with a room temperature probe and processed. Data collection was carried out using excitation sculpting to suppress signals from residual water. Spectra were collected with 1024 transients, 4096-time domain points, and a spectral width of 25,000 Hz.

## Arginylation assays

Arginylation activity was measured in vitro using a modified version of an established mass-spectrometry protocol[66]. Stock RNase-free 4x assay buffer was prepared, comprising 200 mmol/L (4-(2-hydroxyethyl)−1-piperazineethanesulfonic acid) (HEPES), pH 7.5, 100 mmol/L KCl, and 60 mmol/L MgCl$_2$. Stock DTT was added to 0.4 mmol/L final concentration in the 4x assay buffer immediately prior to setting up samples for arginylation assays. The initial reaction mixture was assembled, comprising 0.7 mg/mL *E. coli* or yeast tRNAs, 4 μmol/L *Ec*ArgRS, and 0.5 mmol/L *L*-Arg (all final concentrations) in 1x assay buffer. The initial aminoacylation reaction was initiated by the addition of a solution of Na$_2$ATP to 2.5 mmol/L final concentration. This reaction mixture was incubated at 37 °C with shaking for 2 h to allow for the Arg tRNA$^{Arg}$ to be generated. Afterward, α-lactalbumin (the Arg acceptor protein) and ATE1 were added to 0.125 mg/mL and 2 μmol/L final concentrations, respectively. This reaction mixture was then incubated at 37 °C with shaking for 40 min to allow for the arginylation reaction to take place. The reaction was immediately halted by flash-freezing on N$_{2(l)}$. Prior to analysis, the reaction mixture was desalted via HPLC and then immediately analyzed on a Fourier transform ion cyclotron resonance mass spectrometer. Negative controls lacking the ATE1 construct showed the expected molecular weight of unmodified α-lactalbumin (14176 g/mol (Da)) with no signs of N-terminal modification. SDS-PAGE analysis (acrylamide mass fraction of 15%) on the final reaction mixture revealed no signs of proteolysis of either *Sc*ATE1, *Ec*ArgRS, or α-lactalbumin proteins after the reaction was completed.

## Bioinformatics

All ATE1 sequences were obtained from the Universal Protein Resource (UniProt) Knowledgebase and Reference Clusters (http://www.unprot.org). Sequences were retrieved by standard protein–protein BLAST searches (blastp) using *Sc*ATE1 as an input. Conserved amino acids were identified via multiple sequence alignments that were performed using JalView (version 2.7)[92] implementing the ClustalW algorithm (version 2.1) implementing the Blosum62 matrix[93,94].

## Reporting summary

Further information on research design is available in the Nature Portfolio Reporting Summary linked to this article.

## Data availability

Sourced data are provided with this paper. The data generated in this study are provided in the Supplementary Information/Source Data file. The following Uniprot IDs were used in this study: *Homo sapiens* ATE1, A0A8I5KT53; *Pan troglodytes* ATE1, H2Q2P4; *Canus familiaris* ATE1, A0A8I3PC88 and A0A8I3PH57; *Poephila guttata* ATE1, H0ZM06; *Mus musculus* ATE1, Q80YP1 and Q4FCQ6; *Arabidopsis thaliana* ATE1, Q9ZT48 and Q9C776; *Brachydanio rerio* ATE1, G8XPI0; *Aedes aegypti* ATE1, Q178G8; *Drosophila melanogaster* ATE1, O96539; *Caenorhabditis elegans* ATE1, P90914; *S. cerevisiae* ATE1, P16639; *Escherichia coli* maltose-binding protein, P0AEX9; and *Escherichia coli* ArgRS, P11875. All unique biological materials (e.g., expression plasmids) are readily available from the authors upon request.

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

## Acknowledgements

This work was supported by NIH-NIGMS grant R35 GM133497 (A.T.S.), NIH-NIGMS grant T34 GM136497 (N.-E.E.), and NIH-NIGMS grant R01 GM138557 (F.Z.). In addition, this work was financially supported by the Deutsche Forschungsgemeinschaft (DFG) Priority Program "Iron-Sulfur for Life: Cooperative Function of Iron-Sulfur Centers in Assembly, Biosynthesis, Catalysis and Disease" (SPP 1927) IS 1476/4-1 (I.S.); the Chemical Industry Fund Li 196/05 (I.S.) and Hoe 700080 (H.R.); and the German Academic Scholarship Foundation (H.R.), and by the financial support of the National Institute of Standards and Technology (V.A.S.). Sequence searches utilized both database and analysis functions of the Universal Protein Resource (UniProt) Knowledgebase and Reference Clusters (http://www.uniprot.org) and the National Center for Biotechnology Information (http://www.ncbi.nlm.nih.gov/). Use of the Stanford Synchrotron Radiation Lightsource, SLAC National Accelerator Laboratory, is supported by the U.S. Department of Energy, Office of Science, Office of Basic Energy Sciences under Contract No. DE-AC02-76SF00515. The SSRL Structural Molecular Biology Program is supported by the DOE Office of Biological and Environmental Research, and by the National Institutes of Health, and National Institute of General Medical Sciences (including P41 GM103393).

## Author contributions

V.V., J.B.B., C.R.O., H.R., I.M., K.N.C., V.A.S., F.Z., and A.T.S. designed the research; V.V., J.B.B, C.R.O., H.R., I.M.,N.-E.E., T.S.B., V.A.S., K.N.C., F.Z., and A.T.S. performed the research; V.V., J.B.B., C.R.O., H.R., V.A.S., K.N.C., I.S., F.Z., and A.T.S. analyzed the data; and V.V., J.B.B., C.R.O., H.R., I.M., V.A.S., K.N.C., I.S., F.Z., and A.T.S. wrote and edited the paper.

## Competing interests

The authors declare no competing interests.
