## [Peer Review File · Nature Communications]

Iron-sulfur clusters are involved in post-translational arginylationREVIEWER COMMENTS

Reviewer #1 (Remarks to the Author):

In this study, Van and colleagues provide the biochemical characterization of the *Saccharomyces cerevisiae* Ate1p and to a less detailed extent of the murine Ate1. The authors report the presence of an iron-sulfur cluster in the evolutionarily conserved N-terminal domain of the protein, which contains the molecular determinants required for iron-sulfur cluster acquisition and ligation. The results presented also point to the relevance of the cluster for the arginylation activity of *S. cerevisiae* Ate1p based on an in vitro assay.

Overall, this is an interesting study with potential implications for a better understanding of the regulation of the N-degron pathway and signaling molecules that may affect its activity. The conclusions are for the most part supported by the experimental evidence presented (see detailed comments below), and the results were generated with appropriate techniques and discussed in sufficient details.

I recommend publication after the following comments have been addressed.

1. Purified at ambient air oxygen tension, Ate1p (see further down the minor comment about gene and protein name nomenclature) was found to bind a [2Fe-2S]₂⁺ cluster which exhibited increased activity, compared to apo-Ate1p, towards a model substrate, alpha-lactalbumin, in vitro. While a higher order [4Fe-4S]₂⁺ cluster could be formed by anoxic reconstitution or anoxic purification of Ate1p, this cluster, which is likely physiologically relevant in vivo (in intact cells at physiological oxygen tensions), was degraded to a [2Fe-2S]₂⁺ cluster readily upon exposure to oxygen. The activity assay in vitro was never performed with the [4Fe-4S]-Ate1p. Given that the authors discuss the possibility that the oxygen sensitivity of the cluster in Ate1p may constitute a sensing mechanism that could be leveraged to sense oxidative stress within the cell in order to alter arginylation efficacy and/or substrate specificity, the activity assay with purified [4Fe-4S]-Ate1p should be performed and compared to the activities of the [2Fe-2S]- and apo-Ate1p. I suggest combining the reagents and running the assay in the argon recirculated glove box (to which the authors clearly have access) prior to analyzing the target of arginylation by mass spectrometry.

Related to the comment above:

2. If, as proposed, the cluster in Ate1p is posed to sense O₂, NO and even to respond to oxidative stress, one should consider that not only cysteine residues of proteins targeted by arginylation may be oxidized and/or proteins misfolded with the consequential greater need to flag and degrade these dysfunctional polypeptides, but that also the iron-sulfur cluster in Ate1p may be susceptible to oxidative damage and degraded and that loss of the cluster would affect efficient arginylation when this post-translational modification is most needed. In other words, if the arginylation activity of Ate1p positively correlates with the presence of the cluster, as shown in Figs. 4b-d, then how do the authors explain the apparent contradiction that in conditions in which Ate1p activity is most needed the cofactor which is presumed to be essential for function may be lost? I'm trying to be overly critical here, just attempting to rationalize the potential implications of the discoveries reported. In this sense, knowing the activity of the [4Fe-4S]-Ate1p becomes critical, as the protein may exhibit maximal activity in its [2Fe-2S]-bound form (i.e., upon degradation of the cubane [4Fe-4S] cluster to rhombic [2Fe-2S]), thereby explaining how oxidative degradation of the cluster may lead to greater Ate1p activity, as presumably required in conditions of oxidative stress.

3. Is the anaerobic reconstitution with iron and sulfide performed on apo-Ate1p or is instead the [2Fe-2S]₂⁺-Ate1p being reconstituted?

4. After anoxic reconstitution of Ate1p with 4 mole eq. of iron and sulfide, the stoichiometry of the iron indicated approximately 3 eq. of Fe/polypeptide and a UV-vis spectrum consistent with the presence of a [4Fe-4S] cluster. The underestimation of the iron content in the preparation of the purified protein is very typical in cases in which the protein concentration has been determined by BCA or Bradford (see Lanz et al., 2012, Methods Enzymol [1]). To precisely determine protein concentrations avoiding systematic over or under-estimations inherent to the routinely used colorimetric methods that are

based on the absorbance of a standard protein (usually bovine serum albumin), concentrations determined with the method of Bradford should be corrected with respect to the BSA standard by performing amino acid analysis [1]. The results could be quite different and the re-calculated stoichiometry may better reflect the nature of the cluster present.

5. The reduction of the $[2\text{Fe-2S}]^{2+}$ and $[4\text{Fe-4S}]^{2+}$ with sodium dithionite doesn't seem to be very efficient. What is the percentage of the cluster that is being reduced by dithionite? Spin quantitation of the EPR spectra can help addressing this point.

6. The assertion that the murine Ate11B7A can also ligate an iron-sulfur cluster needs to be further validated to be able to expand the conclusion that the presence of the cluster is a common denominator of eukaryotic ATE1s. I am not requesting evidences provided by Mossbauer spectroscopy. EPR, NMR and EXAFS, used to characterize yeast Ate1p, would suffice.

7. The heme-bound Ate1p aggregated and precipitated in the experiments presented in the paper. Do the authors envision the possibility that the C-terminal domain of Ate1p binds heme in vivo and what would be the physiological relevance of this cofactor?

Minor points:

Pag. 7. The authors mention that the identity of the His-tagged Ate1p, expressed and purified in *E. coli*, was verified by western blotting and they refer to the supplementary figure S1 for the results, but no western results are presented in Fig. S1.

Proper nomenclature for gene and encoded protein names should be used throughout the manuscript.

[1] N.D. Lanz, T.L. Grove, C.B. Gogonea, K.H. Lee, C. Krebs, S.J. Booker, RlmN and AtsB as models for the overproduction and characterization of radical SAM proteins, *Methods Enzymol* 516 (2012) 125-52.

Reviewer #2 (Remarks to the Author):

The authors study in some detail the cofactor binding and function of yeast Ate1, an arginyl-tRNA transferase with importance for cellular protein stability according to the N-end rule. Not much is known to date about the molecular function of this class of proteins, despite their importance for cell proteostasis. In an attempt to understand biological properties of this protein in vitro, the authors purified the protein under aerobic conditions and found a bound $[2\text{Fe-2S}]$ cluster rather than the expected heme b, when expression was performed in the presence of the bacterial *isc* operon. The cluster is verified by iron content (why not sulfur?), UV-Vis, EXAFS, XANES but not by the more telling methods EPR or CD (no signals; and Moessbauer was not attempted). The cluster was shown to be sensitive to oxidizing (O_2) and reducing (DT) conditions and is possibly degraded therefore escaping investigation by EPR (a very usual behavior for clusters). Therefore, I would have expected that a Moessbauer spectrum is recorded. The presented experiments make it likely that Ate1 binds a labile $[2\text{Fe-2S}]$ cluster, but the analysis is far from being convincing. The authors then reconstituted Ate1 chemically which yielded precipitates and a bound under-stoichiometric $[4\text{Fe-4S}]$ cluster (again S content was not measured; no convincing EPR or Moessbauer analysis of this sample). Finally, the authors purified Ate1 under anaerobic conditions using a SufR deletion *E. coli* strain to increase the yield of Fe/S cluster. As judged from the UV Vis spectra (Fig. S5, S6 and Fig. 3f,g) the yield of Fe/S cluster per protein was rather low and possibly again under-stoichiometric (also for mouse ATE1; Fig. S11; no 280 nm peak shown). What was the Fe and S content? EPR shows the presence of $[3\text{Fe-4S}]$ and $[4\text{Fe-4S}]$ clusters which are very labile raising the question whether they are inherently labile (due to functional requirement as claimed by the authors or due to artificial production in a heterologous host *E. coli* under FeS-enforcing conditions. Authors may know that under such conditions artificial FeS binding can easily occur and is a well-known artefact of the FeS field).

While the in vitro data may show the presence of (under-stoichiometric amounts of) FeS in *E. coli* expressed and purified yeast Ate1, there are many unusual observations regarding cluster binding (see above) raising questions about the physiological relevance of this cofactor. Most importantly, it remains unclear whether these clusters are generated in the native environment yeast and whether they have any function (compare to doi: 10.1126/science.abi5224). The only (weak) indication for a functional “importance” of FeS in Ate1 is a lactalbumin arginylation assay which showed a 2-fold increase of arginylation catalyzed by [2Fe-2S] containing Ate1 compared to apo-Ate1. Importantly, this data in fact shows that the Fe/S cluster is NOT required for activity, and hence at best may be relevant for regulation. For the [3Fe-4S] / [4Fe-4S] species similar experiments even failed which is worrying because this protein's species was purified under more careful anoxic conditions. How can the authors exclude that subtle differences in protein quality (e.g., differences in folding or oxidation of the holo- and apoforms) may give rise to this subtle (twofold) functional increase? Overall, the manuscript provides preliminary in vitro evidence for a Fe-S cluster binding activity of yeast Ate1 but fails to address whether the cluster is required in vivo in its native environment. The authors did not study the dependence of arginylation on the ISC and CIA systems in yeast. Further, a functional analysis of Cys mutants in yeast (phenotype?), and direct FeS cluster binding in yeast is needed to convince me of what the title of this manuscript claims. Overall, the content of the current version of the manuscript falls short of convincingly showing the presence and function of a Fe/S cluster in Ate1.

Comments:

1. Fig. S1d: At this concentration of bound cofactor a discrimination of heme or FeS clusters is impossible because both have major peaks at ca. 420 nm. The authors may wish to temper their wording in Figure legend.
2. Generally, for all Ate1 preparations the authors need to determine the sulfur content.
3. The CD spectrum of Fig. S2 bottom does NOT reliably demonstrate a [2Fe-2S] cluster binding. This is noise only (max. signal 1 mdeg). In this Figure (top panel) the values should be shown down to 280 nm (protein peak).
4. Page 8, top: The authors measure Fe but not the sulfur content. Why? This is a necessary information.
5. In the UV Vis Figures, e.g., Fig. S5, 6 and Fig. 3 f) the spectra should include the signal down to at least 280 nm.
6. What is the different FeS species in Fig. 3f? This experiment is poorly defined and leaves the reviewer puzzled.
7. The speculation of a function of the “LYR” motif should be removed unless a function for Fe/S cluster insertion into Ate1 is shown experimentally. LYR and its sequence variants are a frequent tripeptide in proteins (and the conservation of the first residue of “LYR” in Ate1 does not even conform to the degenerate sequence motif of LYR).
8. The mouse ATE1 data are, as the authors admit, very preliminary, and at this stage are not ready for publication.
9. The chapter on heme binding (I guess also in the eyes of the authors) shows how difficult it is to analyze heme binding to proteins. As such, I believe that the authors are right in being skeptical about previous studies of heme association to Ate1 proteins. However, the experiments shown are so limited and lack controls (heme binding to other proteins will show that heme sticks to proteins generally and unspecifically) that I personally would not publish such data quality.

Reviewer #3 (Remarks to the Author):

In the manuscript titled “Iron-sulfur clusters are involved in post-translational arginylation”, Dr. Smith and his colleagues have reported interesting findings that yeast (*Saccharomyces cerevisiae*) ATE1 (ScATE1) was recovered an iron-cluster ([Fe-S]) binding protein. Subsequently, the authors were able to demonstrate that the N-terminal region of scATE1 was mainly responsible for its binding to [Fe-S], by similar bacterial expression and recovery procedure. They went on to demonstrate that at least 1B7A, one isoform of proteins that mouse Ate1 gene encodes, also manifested the [Fe-S]-binding property. In attempt to examine the potential effect of [Fe-S]-binding on the R-transferase activity of ATE1s, the authors tested, and found that dithionite-treated

[Fe-S]-ATE1 arginylated lactalbumin more efficiently than the Apo ATE1, suggesting that ATE1 activity was subjected to the regulation by the reduced status of [Fe-S], particularly with [2Fe-2S]₂-favoring the R-transferase activity of ScATE1. Through series of spectroscopic studies, bioinformatics analyses and reconstitution assays, propose that [Fe-S]-binding is an evolutionarily conserved properties of ATE1s from different origins, with potentially regulatory effect on cellular protein arginylation.

Overall, this is a piece of nice biophysics work performed with yeast ATE1 and a mouse isoform of ATE1s expressed and purified from *E. Coli*, particularly strong in using a variety of spectroscopy approaches to identifying the redox states of ATE1-bound [Fe-S] clusters, with glovebox and dithionite or ascorbate as powerful tools to modulate redox states of ATE1-bound [Fe-S] clusters.

However, much more need to be worked out first, as the questions that this work aimed to address and the model proposed are potentially of great significances, and may have profound impact on the future ways to study all pathophysiological aspects of ATE1 and cellular protein arginylation.

Therefore, it would be essential to perform an array of biochemistry and cellular biology experiments to establish that endogenous ATE1s indeed are indeed [Fe-S]-binding enzymes with more clear mechanistic understanding into how the redox states of the [Fe-S] cluster might regulate the R-transferase activity of ATE1.

Major points:

1. The majority, actually almost all, of the work was done with the bacterially expressed scATE1 (and mouse 1B7A). As the authors were already aware of, and did take advantage of, both the abundance and the redox states of the [Fe-S] cluster were substantially different from yeast and even more so with the mammalian cells. It left a huge gap to fill between the observed binding of ATE1 to [Fe-S] and what might take place with ATE1s that seemed to naturally express in eukaryotes.

2. When it comes to the mechanism and validity of proposed regulatory roles of [Fe-S] clusters on ATE1 activity, it would be essential to go far beyond surmising the potential region(s) in ATE that is implicated in [Fe-S]-binding.

- a. When ATE1-bound [Fe-S] clusters were modulated into different redox states, how did it affect scATE1 activity? Did any of the sulfur groups in CxxC and CC change their redox states in response to the oxic, anoxic or chemical treatment? If not, what other mechanism might be acting there? If yes, can a bunch of mutants be created in as such that some scATE1 might be either mimicking one distinct intermediate status of the different [Fe-S] cluster or constitutively insensitive to the redox states of the protein-bound [Fe-S] clusters.

- b. Once the above mutant ATE1s were created, one may test the arginylation status and the homeostasis/ relevant functions of some (at least one) of the protein arginylation substrates, in the context of ATE1 knockout yeast or mammalian cells.

- c. To precisely map the [Fe-S]-binding regions in scATE1 or mouse ATE11B7A, one may have to test whether the C-terminal regions of ATE1 (downstream to the tested N-terminal fragment of scATE1) could also bind to [Fe-S] or entirely free of [Fe-S]-binding?

- d. To access the effect of heme/hemin on ATE1 solubility and activity, it would be necessary to carry out the experiments in conditions (e.g. anoxic conditions, with or without dithionite treatment etc.) tested in the original paper (Hu et al PNAS 2010), as Reference #33 in this manuscript. The observed effect of hemin on protein aggregation alone may not be sufficient to demonstrate that [Fe-S] is more important.

Minor points:

In Figure 1. Since D and E were used to denotate amino acids in panel a, their uses in panel b might cause confusion in readers.

Reviewer #4 (Remarks to the Author):

This is a very exciting paper that takes an important step in our understanding of the biological actions of Ate1, an essential and enigmatic enzyme. The authors discover a novel Ate1 cofactor that modulates its enzymatic activity and likely mediates some of its previously described in vivo functions. I think this is a key paper that definitely merits publication.

I have only a few comments, related mostly to presentation:

1. I find the background given about Ate1 a bit unbalanced. In the introduction, the authors focus almost exclusively on the N-degron work and characterizes arginylation as a branch of the N-degron pathway. They don't really mention degradation-independent functions of Ate1 here, and even have a "conventional" N-end rule figure, giving a strong impression that Ate1 acts exclusively in this pathway. In the discussion, they shift to the other Ate1 targets that haven't been shown to undergo degradation. I wish that the degradation-dependent and -independent functions were discussed more evenly in the manuscript. My suggestion is to add the degradation-independent Ate1 roles to the introduction and Figure 1, and balance their writing in the discussion too, to reflect the multiple facets of this enzyme.

2. In mouse Ate1 (and Ate1 from other higher vertebrates) the first exon is one of those alternatively spliced, and the position and sequence context of the conserved Cys would vary with this splicing. I felt there could be more discussion of this. E.g., would the sequence context of these exons affect the Fe-S cluster binding? Could this possibly underlie some differences in substrate specificity between the Ate1 isoforms? Can the authors test this, by any chance? This could be a nice addition to this manuscript.

3. The isoform naming convention (e.g., 1B7A) makes sense in terms of alternative splicing, but is not uniformly followed in the literature, and the NCBI database refers to these isoforms as Ate1-1 through Ate1-4. The authors should include a note, and possibly a supplementary diagram, reconciling these conventions so that people not deeply immersed in this field could easily understand which isoforms are being discussed.

4. The conserved Cys residues in the Ate1 sequence have been discussed in multiple early papers, especially the CGYC motif at the very beginning (shown to be critical in early work from Cecile Pickart, and highly conserved between species and critical for activity in work from Varshavsky's lab (e.g., Kwon et al, 1999)). The present study gives new substance to these early findings. Even though the authors do cite all the appropriate prior studies, I kind of wished they could put more emphasis on this information. This is a bit of a freeform comment, so the authors should feel free to do what they think is appropriate here.

5. I found it very intriguing that the authors were clued in on the Ate1 binding [2Fe-2S] cluster by observing the brownish color of the preparation. As I understand it, they illustrate it in Fig. 1a, but I could not quite understand what is shown in the figure. A column? A cartridge? The authors should mark this figure more clearly and include enough details in the figure legend to enable readers to appreciate this result.

We thank the reviewers for the careful attention and thoughtful suggestions (typed gray, below) to our manuscript. Responses to each reviewer (typed in black, below) are enumerated below.

Reviewer 1

We thank this reviewer for their enthusiasm of our work and the recommendation of publication after addressing the following points. We have added additional experimental evidence to support the results of our conclusions, discussed below.

1. *The activity assay in vitro was never performed with the [4Fe-4S]-Ate1p. Given that the authors discuss the possibility that the oxygen sensitivity of the cluster in Ate1p may constitute a sensing mechanism that could be leveraged to sense oxidative stress within the cell in order to alter arginylation efficacy and/or substrate specificity, the activity assay with purified [4Fe-4S]-Ate1p should be performed and compared to the activities of the [2Fe-2S]- and apo-Ate1p.*

2. *If, as proposed, the cluster in Ate1p is posed to sense O₂, NO and even to respond to oxidative stress, one should consider that not only cysteine residues of proteins targeted by arginylation may be oxidized and/or proteins misfolded with the consequential greater need to flag and degrade these dysfunctional polypeptides, but that also the iron-sulfur cluster in Ate1p may be susceptible to oxidative damage and degraded and that loss of the cluster would affect efficient arginylation when this post-translational modification is most needed. In other words, if the arginylation activity of Ate1p positively correlates with the presence of the cluster, as shown in Figs. 4b-d, then how do the authors explain the apparent contradiction that in conditions in which Ate1p activity is most needed the cofactor which is presumed to be essential for function may be lost? I'm not trying to be overly critical here, just attempting to rationalize the potential implications of the discoveries reported. In this sense, knowing the activity of the [4Fe-4S]-Ate1p becomes critical, as the protein may exhibit maximal activity in its [2Fe-2S]-bound form (i.e., upon degradation of the cubane [4Fe-4S] cluster to rhombic [2Fe-2S]), thereby explaining how oxidative degradation of the cluster may lead to greater Ate1p activity, as presumably required in conditions of oxidative stress.*

We appreciate the suggestion of this reviewer to determine the activity of the [4Fe-4S] cluster-bound form of ATE1. After several rounds of optimization in order to ensure a) that the [4Fe-4S] cluster did not degrade in the presence of the additional arginylation components (a non-trivial process as every component is produced under oxic conditions), and b) that the tRNA retained its integrity under anoxic condition after the protein had been reconstituted, we repeatedly tested for ATE1-catalyzed arginylation under anoxic conditions. Notably, we did not observe any arginylation of the mock protein substrate (ALA) under these conditions using our MS-based approach, which would be wholly consistent with our hypothesis posed in the manuscript and that presented by this reviewer. *However*, we were also unable to observe any free ALA in solution, unlike the case of the apo and the [2Fe-2S]-bound forms of ATE1, even after weeks of troubleshooting. The separation of ALA (modified or unmodified) from ATE1 is essential for the top-down MS approach that we developed. As of right now, we strongly believe that the protein substrate is not released from ATE1 when the enzyme is bound to the [4Fe-4S] cluster. Unfortunately, without a structure of the enzyme, and without extensive further probing of the system (requiring additional months of testing), we hesitate to include these data in the revised manuscript. We intend to follow-up on this fascinating turn of events in another work in which we can delve into the molecular-level details of this process.

Regarding the latter point, ALA is mock substrate for our arginylation assays. In the cell, however, there are dozens of *bona fide* intracellular substrates that can be the target of ATE1-catalyzed arginylation (see Wong *et al.*, *PLoS Biology*, **5**, e258 (2007)). In the response of oxidative stress, any of these substrates could have an altered arginylation propensity due to conformational and dynamical changes occurring in ATE1, due to altered binding affinities of these proteins to ATE1, or even due to the presence of now oxidized Cys-sulfinic acids on the N-terminus of these proteins for which ATE1 could be more selective. In truth, there is no current consensus on the exact mechanism by which ATE1 selects its substrates, especially given the lack of structural information on this enzyme. Indeed, we and others are probing further into the selectivity mechanism of this enzyme, but the details are still unclear and far beyond the scope of this manuscript. Despite this roadblock, our conclusion of the manuscript remains true: that [Fe-S] clusters are a novel component of post-translational arginylation, which opens exciting future avenues of exploration in the mechanistic selectivity of this enzyme.

3. Is the anaerobic reconstitution with iron and sulfide performed on apo-Ate1p or is instead the [2Fe-2S]₂⁺-Ate1p being reconstituted?

We have added text to p. 11 to clarify that the anaerobic reconstitution was performed with apo ScATE1.

4. After anoxic reconstitution of Ate1p with 4 mole eq. of iron and sulfide, the stoichiometry of the iron indicated approximately 3 eq. of Fe/polypeptide and a UV-vis spectrum consistent with the presence of a [4Fe-4S] cluster. The underestimation of the iron content in the preparation of the purified protein is very typical in cases in which the protein concentration has been determined by BCA or Bradford (see Lanz et al., 2012, Methods Enzymol]). To precisely determine protein concentrations avoiding systematic over or under-estimations inherent to the routinely used colorimetric methods that are based on the absorbance of a standard protein (usually bovine serum albumin), concentrations determined with the method of Bradford should be corrected with respect to the BSA standard by performing amino acid analysis. The results could be quite different and the re-calculated stoichiometry may better reflect the nature of the cluster present.

We appreciate this reviewer's constructive suggestion to recalibrate our protein concentration based on amino acid analysis. We have done the analysis and recalibrated our protein concentration based on amino acid quantitation. The correction factor based on amino acid quantitation determined was 1.548x. Our updated stoichiometries, which are wholly consistent with our spectroscopic analyses, are now presented throughout the manuscript where we had previously used a Lowry assay to determine protein quantitation.

5. The reduction of the [2Fe-2S]₂⁺ and [4Fe-4S]₂⁺ with sodium dithionite doesn't seem to be very efficient. What is the percentage of the cluster that is being reduced by dithionite? Spin quantitation of the EPR spectra can help addressing this point.

We agree with this reviewer that reduction of *any form* of the cluster bound to the yeast protein is not efficient, and we believe this phenomenon is due to reductive destabilization of the cluster. In fact, in the case of the [2Fe-2S] cluster, reduction appears to cause dissociation of the cluster (see pp. 10-11); due to the lack of antiferromagnetic coupling in this system, there is no resultant S=1/2 species, prohibiting the quantitation. In the case of the anaerobically-purified ScATE1, there are multiple species present depending on the presence/absence of reductant (both [4Fe-4S] and [3Fe-4S], see pp. 13-14 and pp. 22-23); because we know that reduction of the cluster results in destabilization, and we will not be able to compare across all cluster species, we do not believe spin quantitation here will give any meaningful insight into the function of the labile cluster present in ScATE1.

6. The assertion that the murine Ate11B7A can also ligate an iron-sulfur cluster needs to be further validated to be able to expand the conclusion that the presence of the cluster is a common denominator of eukaryotic ATE1s.

In addition to our previous electronic absorption data, we have now prepared and analyzed the mouse ATE1^{1B7A} isoform (now referred to as *MmATE1-1*) with electronic absorption, EPR and X-ray absorption spectroscopies. Consistent with the yeast data, anoxic reconstitution of mouse ATE1-1 displays EPR and XAS consistent with a [4Fe-4S] cluster. Fascinatingly, exposure of the anoxically-reconstituted mouse ATE1-1 to O₂ and subsequent desalting of excess iron allowed us to capture both the [3Fe-4S] and the [2Fe-2S] forms, as judged by EPR spectroscopy and supported by XAS. Thus, not only do these experiments confirm our hypothesis regarding the O₂ sensitivity and mutability of the [Fe-S] cluster based initially on the yeast data, but these data also demonstrate that a cluster is a common cofactor of eukaryotic ATE1s. These data are now presented as Fig. S15 of our revision.

7. The heme-bound Ate1p aggregated and precipitated in the experiments presented in the paper. Do the authors envision the possibility that the C-terminal domain of Ate1p binds heme *in vivo* and what would be the physiological relevance of this cofactor?

We apologize for this misunderstanding, but we do not believe there is any relevance to the spurious heme binding. To clarify: heme-bound ScATE1 only precipitates when excess hemin is titrated into the protein. The protein remains soluble (albeit aggregated) when 1-2 mol. eq. are titrated into the protein. We believe that heme binding is merely an artifact. In the Discussion on pp. 27-28, we do not believe that there is any physiological relevance to the non-

specific binding of heme to ATE1, especially given the spectral similarity of “heme-bound” ScATE1 to that “heme-bound” proteins that bind heme non-specifically:

“Contrary to previous work, we find that ATE1 appears to bind heme non-specifically and induces protein aggregation leading to inactivation, suggesting the heme-based inactivation may be irrelevant to ATE1’s native function.”

Minor points:

Pag. 7. The authors mention that the identity of the His-tagged Ate1p, expressed and purified in E. coli, was verified by western blotting and they refer to the supplementary figure S1 for the results, but no western results are presented in Fig. S1.

We apologize for this oversight and thank the reviewer for pointing out our missing Western blot! This blot is now included in Figure S1.

Proper nomenclature for gene and encoded protein names should be used throughout the manuscript.

We apologize for some differences between how ATE1 is referred to throughout the literature, although we are not entirely sure to which nomenclature the reviewer is referring. As far as we are aware, Ate1p is no longer common nomenclature for yeast ATE1. As we discuss not only multiple forms of ATE1 from several organisms, but also splicing-derived isoforms, we have tried to use organismal abbreviations to clarify all organismal origins of ATE1 throughout (e.g., *Sc* for *Saccharomyces cerevisiae* and *Mm* for *Mus musculus*). However, we believe this reviewer is referring to alternative splicing of isoforms. We have also edited the manuscript to remove previous references to splicing-derived isoforms (e.g., 1B7A), but instead now refer to the specific isoforms ATE1-1 through ATE1-4 (as requested by reviewer 4, *vide infra*).

Reviewer 2

We believe this reviewer was less enthusiastic about our study and requested three general means of improvement of our work: a) further characterization of the cluster-bound forms, more inclusion of protein absorption features throughout, additional characterization of the mouse ATE1, and Mössbauer analyses; b) a demonstration that the cluster is operative in the yeast environment; and c) removal of speculative language about the involvement of the LYR motif in the N-terminal domain and its potential involvement in engaging iron-sulfur cluster biogenesis machinery throughout.

In general, we have addressed the majority of the point-by-point issues below. However:

1) We have not added Mössbauer analyses to our revision, as we have no access to this instrumentation, and we have been especially limited to access of instrumentation at other institutions due to COVID-19 restrictions. Importantly, we can extract the bulk of this necessary information from our XAS studies. With EXAFS analyses, we can probe all of the iron present in the sample, extract average oxidation states (even for several of the EPR-silent samples), correlate cluster composition and the electronic absorption data, and get structural information regarding Fe:S stoichiometries, Fe-S distances, and even Fe-Fe distances, which are *not* directly available via Mössbauer.

Examples of the use of EXAFS for [Fe-S] characterization (both with and without additional spectroscopies) are found in the literature and used by expert groups in the field (see *e.g.*, Broderick and coworkers *J. Biol. Inorg. Chem.*, **19**, 465-483 (2014); Beinert, Thomson, and coworkers *PNAS*, **80**, 393-396 (1983)) and even in our own work (*Biochemistry*, **58**, 4935-4949 (2019)). Thus, we feel that we have very extensive characterization of our various cluster-bound forms.

2) Because we extract and provide Fe:S stoichiometries directly from our EXAFS data, we have not provided additional sulfide quantitation for multiple reasons. First, to do so would require regenerating samples from every single tested preparation throughout the manuscript, and large quantities of protein would need to be consumed and destroyed in order to determine sulfide quantitation through the Beinert *et al.* methodology (*Anal. Biochem.*, **131**, 373-378 (1983)). Second, in addition, the EXAFS data provide *only* the sulfide associated with Fe:S interactions, whereas the Beinert method quantitates *all* sulfide, including any adventitious sulfide not completely removed via buffer exchanging. Finally, it is not very common even for expert labs in the field to report these values, and Fe:S stoichiometries are determined by other methods (see *e.g.*, Krebs, Booker and coworkers *JACS*, **142**, 1911-1924 (2020); Barondeau and coworkers *Biochemistry*, **54**, 3871-3879 (2015); Rouault, Krebs, Bollinger Jr. and coworkers *Science*, **373**, 236-241, (2021); LeBrun and coworkers *Sci. Rep.*, **6**, 31597 (2016); and Broderick and coworkers *Biochemistry*, **56**, 3234-3247 (2017)).

Specific points brought up by this reviewer are discussed below.

1. *Fig. S1d: At this concentration of bound cofactor a discrimination of heme or FeS clusters is impossible because both have major peaks at ca. 420 nm. The authors may wish to temper their wording in Figure legend.*

We have modified the wording in the Fig. S1 legend to read "...The electronic absorption spectrum of this purified protein suggests that [Fe-S] clusters are present at low quantities under these expression and purification conditions after size-exclusion chromatography."

2. *Generally, for all Ate1 preparations the authors need to determine the sulfur content*

Please see the general comment above. In short, this approach is not common and unnecessary as we derive Fe:S stoichiometries directly from our EXAFS data.

3. *The CD spectrum of Fig. S2 bottom does NOT reliably demonstrate a [2Fe-2S] cluster binding. This is noise only (max. signal 1 mdeg). In this Figure (top panel) the values should be shown down to 280 nm (protein peak).*

We disagree that these data represent only noise. While there is clearly noise in the spectrum at 650-800 nm (the weakest portion of the visible absorption spectrum), the remaining portion of the CD spectrum clearly shows multiple negative features and at least one positive, weak feature from 400-650 nm. All data have been buffer-corrected, and these features are not present in the buffer alone. It is true that, in general, CD spectra of [Fe-S] cluster-bound proteins can be very weak, but these spectra are also wholly dependent on 1) the protein concentration, 2) the path length (here only 0.3 cm) and (importantly) 3) the local, structured environment of the polypeptide, which lowers the symmetry of the complex and imparts the chirality on the cluster. Because there is no crystal structure of ATE1, we have no information regarding how structured/unstructured and buried/solvent-exposed the portion of N-terminal domain is where this cluster binds, and as such we cannot predict whether the signal should be strong or weak. We have modified our wording in the Fig. S2 legend to read "The electronic absorption spectrum (top) and circular dichroism (CD) spectrum (bottom) of ScATE1 expressed in the presence of L-Cys, ferric citrate, and iron-sulfur cluster (ISC) biosynthesis machinery, and subsequently purified under oxidic conditions. 0.001° = 1 mdeg."

We are happy to provide the absorption data to 280 nm in the top panel (Fig. S3A) as an inset. Because the near-UV CD spectrum (ca. 190-250 nm) has 2° structural signals that are significantly stronger than those of the visible CD signals from the cluster, we chose to add this as an inset to prevent complication of the figure.

4. *Page 8, top: The authors measure Fe but not the sulfur content. Why? This is a necessary information.*

Please see response to point 2 and our general response above. In short, this approach is not common and unnecessary as we derive Fe:S stoichiometries directly from our EXAFS data.

5. *In the UV Vis Figures, e.g., Fig. S5, 6 and Fig. 3 f) the spectra should include the signal down to at least 280 nm.*

We have expanded the requested spectra to include the signal down to 280 nm as requested for nearly all spectra. The only exceptions to this modification are for Fig. S4, as the addition of sodium dithionite saturates the detector of the instrument at <350 nm, and Figs. S6 and S7, as these spectra are taken from the exact samples we used for EPR analyses ([ScATE1] ca. 200 μ M), and at these concentrations, absorption <300 nm saturates the detector of the instrument. For these samples one cannot extract meaningful information in UV and near-UV absorption windows.

6. *What is the different FeS species in Fig. 3f? This experiment is poorly defined and leaves the reviewer puzzled.*

We apologize for the confusion with the experiment. Our initial figure was meant to show the reactivity of the reconstituted [4Fe-4S]²⁺ ATE1 (yellow) with O₂ to yield the [2Fe-2S]²⁺ ATE1 (purple). The dashed lines displayed the changes in the electronic absorption spectra over a 10-minute reaction period. To simplify this image, we removed the time-course spectra (dashed lines) and display just the starting and ending spectra. As requested in point 5 (*vide supra*), the inset now also shows the full spectra to 280 nm.

7. *The speculation of a function of the "LYR" motif should be removed unless a function for Fe/S cluster insertion into Ate1 is shown experimentally. LYR and its sequence variants are a frequent tripeptide in proteins (and the conservation of the first residue of "LYR" in Ate1 does not even conform to the degenerate sequence motif of LYR).*

We believe that the experimental demonstration of the modified LYK-like motif's involvement in [Fe-S] cluster insertion is well beyond the scope of this manuscript. Although we look forward to probing this speculation further in the future, we have removed the mention of the LYK-motif from this revision, and we have modified Figs. S8 and S9 to reflect only the conservation of the CxxC and CC motifs in the N-terminal domain of ATE1.

8. *The mouse ATE1 data are, as the authors admit, very preliminary, and at this stage are not ready for publication.*

In agreement with this reviewer and reviewer 1, we have added additional spectroscopic characterizations of the mouse ATE1 such as EPR and XAS spectroscopies. Consistent with the yeast data, anoxic reconstitution of mouse ATE1-1 displays EPR and XAS consistent with a [4Fe-4S] cluster. Fascinatingly, exposure of the anoxically-reconstituted mouse ATE1-1 to O₂ and subsequent desalting of excess iron allowed us to capture both the [3Fe-4S] and the [2Fe-2S] forms, as judged by EPR spectroscopy and supported by XAS. Thus, not only do these experiments confirm our hypothesis regarding the O₂ sensitivity and mutability of the [Fe-S] cluster based initially on the yeast data, but these data also demonstrate that a cluster is a common cofactor of eukaryotic ATE1s. These data are now presented as Fig. S15 of our revision.

9. *The chapter on heme binding (I guess also in the eyes of the authors) shows how difficult it is to analyze heme binding to proteins. As such, I believe that the authors are right in being skeptical about previous studies of heme association to Ate1 proteins. However, the experiments shown are so limited and lack controls (heme binding to other proteins will show that heme sticks to proteins generally and unspecifically) that I personally would not publish such data quality.*

We strongly disagree with this assertion that our experiments shown are limited and lack controls. It is well established in the literature that heme can (and indeed does) bind non-specifically to a host of proteins, especially if those proteins have surface-exposed Lewis basic sites such as His and Cys residues or open cavities.

For brevity's sake, we only cited one well-known example in our text from Crane and coworkers (see ref. 64 in our initial submission: the mammalian circadian clock protein Per2: Airola, M. V., Du, J., Dawson, J. H. & Crane, B. R. *Biochemistry*, **49**, 4327-4338 (2010)). Just to list a few additional examples (although there are several more) that either bind heme non-specifically, or may bind non-specifically, include: β -amyloid: Gout, J.; Meuris, F.; Desbois, A.; & Dorlet, P. *J. Inorg. Biochem.* DOI: 10.1016/j.jinorgbio.2021.111664 (2021); the biosynthetic β -barrel protein AbmU: Li, Q.; Ding, W.; Tu, J.; Chi, C.; Huang, H.; Ji, X.; Yao, Z.; Ma, M.; & Ju, J. *ACS Omega*, **5**, 20548-20557 (2020); BSA and several plant proteins in plant lysates: Espinas, N. A.; Kobayashi, K.; Takahashi, S.; Mochizuki,

N.; & Masuda, T. *Plant Cell Physiol.* **53**, 1344-1354 (2012); GAPDH: Hannibal, H.; Collins, D.; Brassard, J.; Chakravarti, R.; Vempati, R.; Dorlet, P.; Santolini, J.; Dawson, J. H.; & Stuehr, D. J. *Biochemistry*, **51**, 8514-8529 (2012). In comparison to these proteins, our data very clearly show spectral properties indicating non-specific interactions between heme and ATE1 that result in protein aggregation and deactivation. We feel very strongly that these experiments are focused rather than limited; given previous assertions that ATE1 was a hemoprotein, these focused experiments were in fact very necessary. As such, we have chosen to keep these data as presented.

Reviewer 3

We appreciate that this reviewer lauded our strong biophysical evidence, was very positive regarding the presentation of the manuscript, and recognized the profound impact that implications from this work would have on future studies of ATE1. We believe that the major criticisms of this reviewer fell into two camps: a) some misinterpretations of the data we have presented (that we have attempted to clarify through textual revisions) and b) requests for additional biochemical and cellular data.

Specific points brought up by this reviewer are discussed below.

1. *The majority, actually almost all, of the work was done with the bacterially expressed scATE1 (and mouse 1B7A). As the authors were already aware of, and did take advantage of, both the abundance and the redox states of the [Fe-S] cluster were substantially different from yeast and even more so with the mammalian cells. It left a huge gap to fill between the observed binding of ATE1 to [Fe-S] and what might take place with ATE1s that seemed to naturally express in eukaryotes.*

While it is true that the majority of our work here was done using recombinant protein expressed heterologously in bacteria, this approach was taken in order to obtain the substantially large quantities of the protein necessary to be able to perform the spectroscopic measurements (typically concentrations necessary are 0.1 – 1 mM). However, in order to arrive at cluster-replete protein, we used the apo form of the protein after purification from the heterologous host and then chemically reconstituted the protein. This approach allows us to generate the cluster-bound forms independent of the expression system of choice. In yeast, we cannot do an overexpression approach, as the overexpression of ATE1 is stress-inducing and pro-apoptotic and will kill the yeast prematurely (see below).

That said, we believe the crux of this reviewer's point is important, and an *in vivo* approach to test the effects of the cluster in a eukaryotic organism (*S. cerevisiae*) is described below (see the response below regarding the Cys→Ser variants).

2. *When it comes to the mechanism and validity of proposed regulatory roles of [Fe-S] clusters on ATE1 activity, it would be essential to go far beyond surmising the potential region(s) in ATE that is implicated in [Fe-S]-binding.*

a. *When ATE1-bound [Fe-S] clusters were modulated into different redox states, how did it affect scATE1 activity? Did any of the sulfur groups in CxxC and CC change their redox states in response to the oxic, anoxic or chemical treatment? If not, what other mechanism might be acting there? If yes, can a bunch of mutants be created in as such that some scATE1 might be either mimicking one distinct intermediate status of the different [Fe-S] cluster or constitutively insensitive to the redox states of the protein-bound [Fe-S] clusters.*

We apologize for the confusion here, but we believe this point is a result of a misinterpretation or a misunderstanding of the general way that [Fe-S] clusters bind to proteins. As the redox state of the cluster changes, this is (in general) a change in the oxidation of the *iron* ion, not the thiolate side chains of the Cys residues. There are some examples in which the cluster decomposes upon *sulfide* insertion during substrate transformation (see *e.g.*, LipA or biotin synthase, Drennan, Booker, and coworkers), but even those enzymes conform to the canonical view of the field that the Cys residues in [Fe-S] clusters remain coordinated to their iron ions as Cys(thiolate) ligands while the metal cycles throughout its oxidation state. However, the latter point (Cys→Ser variants) is interesting, and we describe our approach of answering this question below.

b. Once the above mutant ATE1s were created, one may test the arginylation status and the homeostasis/ relevant functions of some (at least one) of the protein arginylation substrates, in the context of ATE1 knockout yeast or mammalian cells.

We have now created a suite of Cys→Ser variant proteins in which we have mutated our recombinant system to change the CxxC motif to SxxS, the CC motif to SS, and then the combination of the two. As expected, and consistent with our previous hypothesis, recombinant Cys→Ser variant ScATE1 proteins show impaired cluster binding, with the CxxC→SxxS variation showing a more dramatic impact on the cluster coordination than the CC→SS variation, although both impact iron loading and the final spectral features. Even in the quadruple variant, we still see some adventitious binding, but the quantitation and spectral features are dramatically diminished and altered. These results strongly implicate and importance in these residues as essential for cluster coordination (see pp. 16-17 in our revised manuscript and the new Supplementary Figure 10).

We then went further and performed *in vivo* mutations to test how/whether the Cys→Ser substitutions in yeast affect ATE1 function in the cell and result in an altered phenotype. In order to demonstrate the importance of cluster binding to the yeast ATE1 in the parent organism, we have now transiently expressed variant forms of ATE1. As it is well known that overexpression of functional ATE1 sensitizes yeast to heat stress and is ultimately pro-apoptotic, knock-out of yeast ATE1 and transient expression of a non-functional ATE1 in yeast rescues this phenotype. Consistent with our findings, doing so with CxxC→SxxS -GFP and His-tagged versions of variant ATE1s alters the yeast stress response, as would be expected for a non-functional or compromised ATE1. The same is true for the CC→SS -GFP and His-tagged ATE1 variants that are transiently expressed. Combined with our *in vitro* data, these *in vivo* findings strongly support that perturbation of the cluster-binding residues alters function of ATE1 in the cell. These important results are described in our revised manuscript (see pp. 20-21).

We believe these additional data not only demonstrate the importance of these N-terminal Cys residues in cluster coordination, but also the *in vivo* relevance of cluster binding to yeast homeostasis.

c. To precisely map the [Fe-S]-binding regions in scATE1 or mouse ATE11B7A, one may have to test whether the C-terminal regions of ATE1 (downstream to the tested N-terminal fragment of scATE1) could also bind to [Fe-S] or entirely free of [Fe-S]-binding?

We disagree that these experiments are necessary, and we also have evidence that these experiments may not be feasible. Given that 1) our N-terminal fusion shows [Fe-S] cluster binding virtually identically to WT ScATE1, that 2) mutation of these N-terminal Cys residues shows no or diminished [Fe-S] cluster binding, that 3) mutation of these residues *in vivo* demonstrates a yeast phenotype, that 4) previous studies by Pickart and coworkers (*Biochemistry*, **34**, 139-147 (1995)) have shown an important role for these Cys residues for *in vitro* activity, and that 5) these four Cys residues are conserved from yeast to humans in the N-terminal domain, these data unequivocally point to the N-terminal domain as the [Fe-S] cluster binding site. In addition, we have cloned a C-terminal fragment of ScATE1, and despite exhaustive trials, this fragment failed to accumulate. We are uncertain why (unstable expression, poor folding, toxicity), but it does not seem feasible nor necessary at this moment to express only a C-terminal fragment to generate a negative result.

d. To access the effect of heme/hemin on ATE1 solubility and activity, it would be necessary to carry out the experiments in conditions (e.g. anoxic conditions, with or without dithionite treatment etc.) tested in the original paper (Hu et al PNAS 2010), as Reference #33 in this manuscript. The observed effect of hemin on protein aggregation alone may not be sufficient to demonstrate that [Fe-S] is more important.

We believe that we have already faithfully recapitulated the essential aspects of the experiments performed by Hu et al. (PNAS, 2010), which demonstrates an effect on the activity of ATE1 upon titration of hemin into the protein that we interpret as non-specific. As far as we are aware, the Hu et al. manuscript (PNAS, 2010) never actually demonstrates that reduced, ferrous heme interacts with ATE1, only oxidized, ferric hemin. Unfortunately, we cannot perform anoxic size-exclusion chromatography on ferrous heme-ATE1, as we do not have the ability to do so under strictly anoxic conditions and still perform the multi-wavelength co-elution analysis that we reported in our original manuscript (that instrumentation is only available under oxic conditions). Given that the initial manuscript never

reported that ferrous heme could interact with ATE1 (it may not interact at all, and there is no evidence that it does), we do not find a compelling argument to test it here. Finally, in our manuscript, we do not argue which is more important; rather, we present our data and the previous data to suggest that the interaction of hemin with ATE1 may be an off-target or non-specific effect (see pp. 27-28 of our revision). It is possible that, under non homeostatic conditions, hemin may interact with ATE1 and elicit an effect, but we believe this effect is likely to be non-specific.

Minor points: In Figure 1. Since D and E were used to denote amino acids in panel a, their uses in panel b might cause confusion in readers.

We appreciate pointing out this potentially confusing aspect to our previous version of Fig. 1. We have now modified Fig. 1 so that it only includes the “R” label indicative of the arginylated amino acid.

Reviewer 4

We appreciate this reviewer’s recognition of the importance of our findings as they relate to ATE1 function, and that our results were meritorious of publication. We believe we have addressed the bulk of this reviewer’s thoughtful suggestions to help improve our presentation (described below).

1. I find the background given about Ate1 a bit unbalanced. In the introduction, the authors focus almost exclusively on the N-degron work and characterizes arginylation as a branch of the N-degron pathway. They don’t really mention degradation-independent functions of Ate1 here, and even have a “conventional” N-end rule figure, giving a strong impression that Ate1 acts exclusively in this pathway. In the discussion, they shift to the other Ate1 targets that haven’t been shown to undergo degradation. I wish that the degradation-dependent and -independent functions were discussed more evenly in the manuscript. My suggestion is to add the degradation-independent Ate1 roles to the introduction and Figure 1, and balance their writing in the discussion too, to reflect the multiple facets of this enzyme.

We agree with this reviewer that both the N-degron aspects of arginylation as well as the regulatory aspects of arginylation are equally important. To balance the manuscript further, we have now modified the Introduction, and portions of Figure 1 to attempt to strike a stronger balance between degradation-dependent and -independent functions. However, because the N-degron pathway is very complicated, a certain amount of explanation behind the N-degron pathway must remain in the Introduction to be fully understood by the reader. Moreover, we believe our initial Discussion section on this topic struck an appropriate balance between the two, so we have chosen not to change this section.

2. In mouse Ate1 (and Ate1 from other higher vertebrates) the first exon is one of those alternatively spliced, and the position and sequence context of the conserved Cys would vary with this splicing. I felt there could be more discussion of this. E.g., would the sequence context of these exons affect the Fe-S cluster binding? Could this possibly underlie some differences in substrate specificity between the Ate1 isoforms? Can the authors test this, by any chance? This could be a nice addition to this manuscript.

We agree with this reviewer that it is both an interesting and an intriguing possibility that the different isoforms of ATE1 in higher-order organisms could potentially alter the behavior of ATE1 towards [Fe-S] cluster binding and reactivity. However, the bulk of this manuscript focuses on yeast ATE1, which only has a single isoform and which greatly simplifies the extensive analyses necessary. While we do show that mouse ATE1^{1B7A} (ATE1-1) binds an oxygen-reactive [Fe-S] cluster, the *in vitro* and *in vivo* testing of the alternatively spliced isoforms and their reactivity lie outside the scope and the extent of this manuscript. Nevertheless, we look forward to testing this hypothesis in the future, and we appreciate the suggestion.

3. The isoform naming convention (e.g., 1B7A) makes sense in terms of alternative splicing, but is not uniformly followed in the literature, and the NCBI database refers to these isoforms as Ate1-1 through Ate1-4. The authors should include a note, and possibly a supplementary diagram, reconciling these conventions so that people not deeply immersed in this field could easily understand which isoforms are being discussed.

This reviewer brings up an important point that was also caught by reviewer 1. We have changed our nomenclature for *MmATE1-1* to follow the NCBI database nomenclature. We have also added additional text to clarify in the legend of an additional supplementary figure (Supplemental Figure 1 in our resubmission).

4. The conserved Cys residues in the Ate1 sequence have been discussed in multiple early papers, especially the CGYC motif at the very beginning (shown to be critical in early work from Cecile Pickart, and highly conserved between species and critical for activity in work from Varshavsky's lab (e.g., Kwon et al, 1999)). The present study gives new substance to these early findings. Even though the authors do cite all the appropriate prior studies, I kind of wished they could put more emphasis on this information. This is a bit of a freeform comment, so the authors should feel free to do what they think is appropriate here.

We appreciate this reviewer's comment that our work gives more substance and grounding to these earlier findings. As requested, we have tried to emphasize the importance of our findings *vis-à-vis* these earlier studies (see now our revised Discussion pp. 26-27).

5. I found it very intriguing that the authors were clued in on the Ate1 binding [2Fe-2S] cluster by observing the brownish color of the preparation. As I understand it, they illustrate it in Fig. 1a, but I could not quite understand what is shown in the figure. A column? A cartridge? The authors should mark this figure more clearly and include enough details in the figure legend to enable readers to appreciate this result.

We apologize for the confusion. Our initial submission of Fig. 2a illustrated soluble, purified protein collected as eluent in a glass tube from an FPLC-based purification. As this process may not be familiar to some (even many) readers, we have substituted this image for an image of the purified protein contained in a 1.5 mL Eppendorf tube, which is likely more universally familiar to most readers. We have also clarified this point in the Fig. 2a legend.

REVIEWER COMMENTS

Reviewer #1 (Remarks to the Author):

The authors have satisfactorily addressed all the concerns raised during the first round of review. I recommend publication of the manuscript.

I suggest leaving the speculation about the potential function of the "LYR" motif and return to the versions of Figs. S8 and S9 present in the original submission. Including these display items doesn't require a thorough experimental investigation of the role of the LYK motif of ATE1 in iron-sulfur cluster acquisition in the present study.

Reviewer #2 (Remarks to the Author):

Reviewer 2 my response

We believe this reviewer was less enthusiastic about our study and requested three general means of improvement of our work: a) further characterization of the cluster-bound forms, more inclusion of protein absorption features throughout, additional characterization of the mouse ATE1, and Mössbauer analyses; b) a demonstration that the cluster is operative in the yeast environment; and c) removal of speculative language about the involvement of the LYR motif in the N-terminal domain and its potential involvement in engaging iron-sulfur cluster biogenesis machinery throughout. In general, we have addressed the majority of the point-by-point issues below.

Unfortunately, the authors have NOT responded to my major concern to analyze the in vivo relevance of FeS clusters on arginylation in yeast. In my eyes, a needed approach is the examination of ISC and/or CIA defects on arginylation to support the headline claim of the paper. Simply ignoring this point is confusing.

However:

1) We have not added Mössbauer analyses to our revision, as we have no access to this instrumentation, and we have been especially limited to access of instrumentation at other institutions due to COVID-19 restrictions.

I accept the COVID situation. Yet at a later stage, such experiments may well be needed.

Importantly, we can extract the bulk of this necessary information from our XAS studies. With EXAFS analyses, we can probe all of the iron present in the sample, extract average oxidation states (even for several of the EPR-silent samples), correlate cluster composition and the electronic absorption data, and get structural information regarding Fe:S stoichiometries, Fe-S distances, and even Fe-Fe distances, which are not directly available via Mössbauer. Examples of the use of EXAFS for [Fe-S] characterization (both with and without additional spectroscopies) are found in the literature and used by expert groups in the field (see e.g., Broderick and coworkers *J. Biol. Inorg. Chem.*, 19, 465-483 (2014); Beinert, Thomson, and coworkers *PNAS*, 80, 393-396 (1983)) and even in our own work (*Biochemistry*, 58, 4935-4949 (2019)). Thus, we feel that we have very extensive characterization of our various cluster-bound forms.

2) Because we extract and provide Fe:S stoichiometries directly from our EXAFS data, we have not provided additional sulfide quantitation for multiple reasons. First, to do so would require regenerating samples from every single tested preparation throughout the manuscript, and large quantities of protein would need to be consumed and destroyed in order to determine sulfide quantitation through the Beinert et al. methodology (*Anal. Biochem.*, 131, 373-378 (1983)). Second, in addition, the EXAFS data provide only the sulfide associated with Fe:S interactions, whereas the Beinert method quantitates all sulfide, including any adventitious sulfide not completely removed via buffer exchanging. Finally, it is not very common even for expert labs in the field to report these values, and Fe:S stoichiometries are determined by other methods (see e.g., Krebs, Booker and coworkers *JACS*, 142, 1911-1924 (2020); Barondeau and coworkers *Biochemistry*, 54, 3871-3879 (2015));

Rouault, Krebs, Bollinger Jr. and coworkers *Science*, 373, 236-241, (2021); LeBrun and coworkers *Sci. Rep.*, 6, 31597 (2016); and Broderick and coworkers *Biochemistry*, 56, 3234-3247 (2017)).

I respectfully disagree here. Sulfide determination is simple, rapid (2 h) and does NOT need extensive amounts of material (at least in comparison the EXFAS, EPR and Moessbauer). It is a valuable (additional) measure to see whether Fe and S contents agree and whether close to stoichiometric amounts are bound to the protein. For the current study I would have found this helpful, the more so as the spectral data indicate an under-stoichiometric binding of FeS (extinction coefficients are now extractable from the full spectrum UV/Vis data).

Specific points brought up by this reviewer are discussed below.

1. Fig. S1d: At this concentration of bound cofactor a discrimination of heme or FeS clusters is impossible because both have major peaks at ca. 420 nm. The authors may wish to temper their wording in Figure legend.

We have modified the wording in the Fig. S1 legend to read "...The electronic absorption spectrum of this purified protein suggests that [Fe-S] clusters are present at low quantities under these expression and purification conditions after size-exclusion chromatography."

OK.

2. Generally, for all Ate1 preparations the authors need to determine the sulfur content. Please see the general comment above. In short, this approach is not common and unnecessary as we derive Fe:S stoichiometries directly from our EXAFS data.

As written above, I disagree in this general point, since sulfide determinations are quick and easy to do and do not need excessive amounts of material (at least in comparison to EXAFS and EPR requirements). I do not really understand why these simple additions were not provided (even though new mouse data were generated).

3. The CD spectrum of Fig. S2 bottom does NOT reliably demonstrate a [2Fe-2S] cluster binding. This is noise only (max. signal 1 mdeg). In this Figure (top panel) the values should be shown down to 280 nm (protein peak).

We disagree that these data represent only noise. While there is clearly noise in the spectrum at 650-800 nm (the weakest portion of the visible absorption spectrum), the remaining portion of the CD spectrum clearly shows multiple negative features and at least one positive, weak feature from 400-650 nm. All data have been buffer-corrected, and these features are not present in the buffer alone. It is true that, in general, CD spectra of [Fe-S] cluster-bound proteins can be very weak, but these spectra are also wholly dependent on 1) the protein concentration, 2) the path length (here only 0.3 cm) and (importantly) 3) the local, structured environment of the polypeptide, which lowers the symmetry of the complex and imparts the chirality on the cluster. Because there is no crystal structure of ATE1, we have no information regarding how structured/unstructured and buried/solvent-exposed the portion of N-terminal domain is where this cluster binds, and as such we cannot predict whether the signal should be strong or weak. We have modified our wording in the Fig. S2 legend to read "The electronic absorption spectrum (top) and circular dichroism (CD) spectrum (bottom) of ScATE1 expressed in the presence of L-Cys, ferric citrate, and iron-sulfur cluster (ISC) biosynthesis machinery, and subsequently purified under oxic conditions. $0.001^\circ = 1$ mdeg."

We are happy to provide the absorption data to 280 nm in the top panel (Fig. S3A) as an inset. Because the near-UV CD spectrum (ca. 190-250 nm) has 2° structural signals that are significantly stronger than those of the visible CD signals from the cluster, we chose to add this as an inset to prevent complication of the figure.

The newly added insert for UV/Vis is a valuable and necessary information (and clearly shows the under-stoichiometric cluster contents). However, the correct control for the CD signal would be cluster-less protein not a buffer control, e.g., after destroying the cluster or a Cys mutant. Therefore, without new information I am still skeptical about the significance of the weak and noisy CD signal.

4. Page 8, top: The authors measure Fe but not the sulfur content. Why? This is a necessary information.

Please see response to point 2 and our general response above. In short, this approach is not common and unnecessary as we derive Fe:S stoichiometries directly from our EXAFS data.

As mentioned above, I disagree here.

5. In the UV Vis Figures, e.g., Fig. S5, 6 and Fig. 3 f) the spectra should include the signal down to at least 280 nm.

We have expanded the requested spectra to include the signal down to 280 nm as requested for nearly all spectra. The only exceptions to this modification are for Fig. S4, as the addition of sodium dithionite saturates the detector of the instrument at <350 nm, and Figs. S6 and S7, as these spectra are taken from the exact samples we used for EPR analyses ([ScATE1] ca. 200 μ M), and at these concentrations, absorption <300 nm saturates the detector of the instrument. For these samples one cannot extract meaningful information in UV and near-UV absorption windows.

Valuable information was added to some Figs. Thank you. (For Figs. S6 and S7: Why not simply diluting the samples and reading the 280 nm signal?). More important: Why is the 420 nm signal of the SxxS mutant (grey) so high in Fig. S10 (comparable to wild-type)? Why do the authors write: "The largest impact in cluster binding appears to alteration of the C20xxC23 motif to S20xxS23."? Mutation of the Cys should give NO signal at 420 nm, yet the signal is still there with additional absorption below 400 nm. Iron binding is usually at 320 nm. This Figure is confusing me.

6. What is the different FeS species in Fig. 3f? This experiment is poorly defined and leaves the reviewer puzzled.

We apologize for the confusion with the experiment. Our initial figure was meant to show the reactivity of the reconstituted [4Fe-4S]₂⁺ ATE1 (yellow) with O₂ to yield the [2Fe-2S]₂⁺ ATE1 (purple). The dashed lines displayed the changes in the electronic absorption spectra over a 10-minute reaction period. To simplify this image, we removed the time-course spectra (dashed lines) and display just the starting and ending spectra. As requested in point 5 (vide supra), the inset now also shows the full spectra to 280 nm.

Thank you for the clarification and explanation. The 280 nm comparison shows how little FeS cluster is attached to this sample.

7. The speculation of a function of the "LYR" motif should be removed unless a function for Fe/S cluster insertion into Ate1 is shown experimentally. LYR and its sequence variants are a frequent tripeptide in proteins (and the conservation of the first residue of "LYR" in Ate1 does not even conform to the degenerate sequence motif of LYR).

We believe that the experimental demonstration of the modified LYK-like motif's involvement in [Fe-S] cluster insertion is well beyond the scope of this manuscript. Although we look forward to probing this speculation further in the future, we have removed the mention of the LYK-motif from this revision, and we have modified Figs. S8 and S9 to reflect only the conservation of the CxxC and CC motifs in the N-terminal domain of ATE1.

Ok.

8. The mouse ATE1 data are, as the authors admit, very preliminary, and at this stage are not ready for publication.

In agreement with this reviewer and reviewer 1, we have added additional spectroscopic characterizations of the mouse ATE1 such as EPR and XAS spectroscopies. Consistent with the yeast data, anoxic reconstitution of mouse ATE1-1 displays EPR and XAS consistent with a [4Fe-4S] cluster. Fascinatingly, exposure of the anoxically reconstituted mouse ATE1-1 to O₂ and subsequent

desalting of excess iron allowed us to capture both the [3Fe-4S] and the [2Fe-2S] forms, as judged by EPR spectroscopy and supported by XAS. Thus, not only do these experiments confirm our hypothesis regarding the O₂ sensitivity and mutability of the [Fe-S] cluster based initially on the yeast data, but these data also demonstrate that a cluster is a common cofactor of eukaryotic ATE1s. These data are now presented as Fig. S15 of our revision.

This data is a valuable addition and shows that the mouse and yeast proteins behave similarly. However, it also shows how little FeS is bound.

9. The chapter on heme binding (I guess also in the eyes of the authors) shows how difficult it is to analyze heme binding to proteins. As such, I believe that the authors are right in being skeptical about previous studies of heme association to Ate1 proteins. However, the experiments shown are so limited and lack controls (heme binding to other proteins will show that heme sticks to proteins generally and unspecifically) that I personally would not publish such data quality.

We strongly disagree with this assertion that our experiments shown are limited and lack controls. It is well established in the literature that heme can (and indeed does) bind non-specifically to a host of proteins, especially if those proteins have surface-exposed Lewis basic sites such as His and Cys residues or open cavities.

For brevity's sake, we only cited one well-known example in our text from Crane and coworkers (see ref. 64 in our initial submission: the mammalian circadian clock protein Per2: Airola, M. V., Du, J., Dawson, J. H. & Crane, B. R. *Biochemistry*, 49, 4327-4338 (2010)). Just to list a few additional examples (although there are several more) that either bind heme non-specifically, or may bind non-specifically, include: b-amyloid: Gout, J.; Meuris, F.; Desbois, A.; & Dorlet, P. J. *Inorg. Biochem.* DOI: 10.1016/j.jinorgbio.2021.111664 (2021); the biosynthetic b-barrel protein AbmU: Li, Q.; Ding, W.; Tu, J.; Chi, C.; Huang, H.; Ji, X.; Yao, Z.; Ma, M.; & Ju, J. *ACS Omega*, 5, 20548-20557 (2020); BSA and several plant proteins in plant lysates: Espinas, N. A.; Kobayashi, K.; Takahashi, S.; Mochizuki, N.; & Masuda, T. *Plant Cell Physiol.* 53, 1344-1354 (2012); GAPDH: Hannibal, H.; Collins, D.; Brassard, J.; Chakravarti, R.; Vempati, R.; Dorlet, P.; Santolini, J.; Dawson, J. H.; & Stuehr, D. J. *Biochemistry*, 51, 8514-8529 (2012). In comparison to these proteins, our data very clearly show spectral properties indicating non-specific interactions between heme and ATE1 that result in protein aggregation and deactivation. We feel very strongly that these experiments are focused rather than limited; given previous assertions that ATE1 was a hemoprotein, these focused experiments were in fact very necessary. As such, we have chosen to keep these data as presented.

There are no new data added to address my concerns, even though the authors agree with my view that studying heme binding to proteins is a difficult issue.

Reviewer #3

In comments to the editor the reviewer stated that the added work has improved the in vitro aspects of the work but the work has not moved beyond this and it was never demonstrated that ATE1 formed a complex with the proposed [Fe-S] either in yeast or mammalian cells if such clusters exist.

The reviewer points out that the work, in places, is an extension of work reported in REF 38.

The reviewer also states that the evidence disagreeing with the literature view is limited and that evidence to support the literature view continues to be published.

Reviewer #4 (Remarks to the Author):

The authors addressed most of my comments.

Regarding the position of the Cys motif in the different Ate1 isoforms, I was hoping for some insights into sequence alignment, not experiments. I understand that the current work is based on yeast ATE1, but at least some speculations would be helpful.

The manuscript is strangely formatted in my version, with many pages including only two lines or lines broken off in between. Some figure legends don't fit into the text boxes. I assume this will be corrected, but it made it more difficult to evaluate the revised manuscript.

I appreciate replacing the image in Fig. 2 with an Eppendorf tube to demonstrate the brownish color, but it would have been nice to have a comparison here -- an empty tube? A clear solution? something to illustrate the color difference?

We thank the reviewers for the further attentive and thoughtful suggestions (typed gray, below) to our manuscript. Responses to each reviewer (typed in black, below) are enumerated below.

Reviewer 1

1. The authors have satisfactorily addressed all the concerns raised during the first round of review. I recommend publication of the manuscript. I suggest leaving the speculation about the potential function of the "LYR" motif and return to the versions of Figs. S8 and S9 present in the original submission. Including these display items doesn't require a thorough experimental investigation of the role of the LYK motif of ATE1 in iron-sulfur cluster acquisition in the present study.

We appreciate the suggestion and we agree with this reviewer; however, at the request of reviewer 2, we have chosen to keep this portion of the figure removed to simplify the presentation.

Reviewer 2

1. Unfortunately, the authors have NOT responded to my major concern to analyze the *in vivo* relevance of FeS clusters on arginylation in yeast. In my eyes, a needed approach is the examination of ISC and/or CIA defects on arginylation to support the headline claim of the paper. Simply ignoring this point is confusing.

We respectfully accept this critique and apologize that we were unable to deliver these experiments in our previous revision. The disruption of ISC/CIA pathways is not easy because most of the involved enzymes are essential for yeast viability. Fortunately, we were able to identify two yeast strains with *yfh1* or *met18* knockouts, which are known to cause deficiencies in the ISC and/or CIA pathways via different mechanisms. We expressed an established arginylation reporter in these mutant yeasts to measure the *in vivo* arginylation levels. **We found that the arginylation activity was indeed significantly reduced in these yeasts despite expressing similar amounts of ATE1 compared to the WT strain.** Thus, these additional results provide significant and compelling evidence further strengthening our conclusion that [Fe-S] clusters are linked to post-translational arginylation. These new data are presented on pp. 19-22 of our further revised manuscript.

2. I respectfully disagree here. Sulfide determination is simple, rapid (2 h) and does NOT need extensive amounts of material (at least in comparison the EXFAS, EPR and Moessbauer). It is a valuable (additional) measure to see whether Fe and S contents agree and whether close to stoichiometric amounts are bound to the protein.

We have regenerated the key WT ScATE1 samples in the manuscript and quantitated the sulfide amount according to a modified method of Beinert's (*Anal. Biochem.* **131**, 373-378 (1983)) detailed in Ayikpoe, R. *et al. Biochemistry* **58**, 940-950 (2019). Briefly, cluster-containing protein was incubated 1% (w/v) zinc acetate and 7% (w/v) sodium hydroxide and incubated at room temperature for approximately 15 min. After incubation, 0.1% (w/v) *N,N*-dimethyl-*p*-phenylenediamine (dissolved in 5 M HCl) and 10 mM FeCl₃ (stabilized in 1 M HCl) were added, mixed, and the solution was incubated at room temperature for approximately 20 min. The chemically-generated methylene blue was measured at 670 nm with an extinction coefficient (ϵ_{670}) of $\approx 35 \text{ mmol L}^{-1} \text{ cm}^{-1}$, and the concentration of sulfide in the sample was determined.

As expected, we measured ~ 2 Fe per polypeptide and ~ 2 sulfide per polypeptide of the ScATE1 expressed in the presence of the ISC machinery, which binds a [2Fe-2S] cluster. As for the reconstituted protein, the sulfide is slightly sub-stoichiometric: we measured ~ 4 Fe per polypeptide and ~ 3 sulfide per polypeptide of the anoxically-reconstituted ScATE1. This latter result suggests some sub-stoichiometric

loading of the [4Fe-4S] cluster, consistent with our slightly low molar absorptivity (see point 3 below). Nevertheless, our spectral data point to the presence of a [4Fe-4S] cluster. These new data are now presented in our further revised manuscript.

3. For the current study I would have found this helpful, the more so as the spectral data indicate an under-stoichiometric binding of FeS (extinction coefficients are now extractable from the full spectrum UV/Vis data).

We respectfully disagree with this assessment. The yeast ATE1 A₂₈₀ extinction coefficient is $\approx 136,000 \text{ M}^{-1} \text{ cm}^{-1}$ for the holo [4Fe-4S]²⁺ protein based on amino acid analyses requested in the previous round of reviews by reviewer 1.

As just one example, we have graphed Fig. 3a below as molar absorptivity vs. wavelength. For the [4Fe-4S]²⁺ species, ϵ at 405 nm (the peak maximum of the cluster) is $\approx 13,100 \text{ M}^{-1} \text{ cm}^{-1}$; for the [2Fe-2S]²⁺ species, ϵ at 414 nm (the peak maximum of the cluster) is $\approx 19,700 \text{ M}^{-1} \text{ cm}^{-1}$.

These values are sufficiently in line with literature values, which can vary based on maximal wavelength reported; for example:

ϵ [4Fe-4S]²⁺ FNR: $\approx 16,000 \text{ M}^{-1} \text{ cm}^{-1}$ (Thomson, LeBrun, and coworkers; see *JACS*, DOI: 10.1021/ja077455+)

ϵ [4Fe-4S]²⁺ biotin synthase: $\approx 15,000 \text{ M}^{-1} \text{ cm}^{-1}$ (Jarrett and coworkers; see *Biochemistry*, DOI: 10.1021/bi0104625)

ϵ [4Fe-4S]²⁺ WhiD: $\approx 16,000 \text{ M}^{-1} \text{ cm}^{-1}$ (Thomson, LeBrun, and coworkers; see *Biochemistry*, DOI: 10.1021/bi901498v)

ϵ [4Fe-4S]²⁺ FeoC: $\approx 15,000 \text{ M}^{-1} \text{ cm}^{-1}$ (Smith and coworkers; see *Biochemistry*, DOI: 10.1021/acs.biochem.9b00745)

ϵ [4Fe-4S]²⁺ NfuA: $\approx 15,000 \text{ M}^{-1} \text{ cm}^{-1}$ (Johnson and coworkers; see *JBC*, DOI: 10.1074/jbc.M709161200)

ϵ [2Fe-2S]²⁺ RNR: $\approx 16,000 \text{ M}^{-1} \text{ cm}^{-1}$ (Fontecave and coworkers; see *JACS*, DOI: 10.1021/ja990073m)

ϵ [2Fe-2S]²⁺ IscA: $\approx 10,000 \text{ M}^{-1} \text{ cm}^{-1}$ (Johnson and coworkers; see *Biochemistry*, DOI: 10.1021/bi3006658)

ϵ [2Fe-2S]²⁺ ferrochelatase: $\approx 20,000 \text{ M}^{-1} \text{ cm}^{-1}$ (Johnson and coworkers; at $\approx 420 \text{ nm}$; see *Biochemistry*, DOI: 10.1021/bi00168a003)

ϵ [2Fe-2S]²⁺ GLRX₃-GS₄: $\approx 17,000 \text{ M}^{-1} \text{ cm}^{-1}$ (Banci and coworkers; see *JACS*, DOI: 10.1021/jacs.0c02266)

Thus, while there is variability across the literature, our calculated molar absorptivity values indicate significant incorporation of the clusters. Our calculated molar absorptivity of our anoxically-reconstituted ScATE1 is slightly low compared to the average aggregate literature values, but as shown above there is some underloading of the replete [4Fe-4S] cluster based on sulfide quantitation.

4. As written above, I disagree in this general point, since sulfide determinations are quick and easy to do and do not need excessive amounts of material (at least in comparison to EXAFS and EPR requirements). I do not really understand why these simple additions were not provided (even though new mouse data were generated).

We have added the sulfide quantitation of the key samples, as described in point 2.

5. The newly added insert for UV/Vis is a valuable and necessary information (and clearly shows the under-stoichiometric cluster contents). However, the correct control for the CD signal would be cluster-less protein not a buffer control, e.g., after destroying the cluster or a Cys mutant. Therefore, without new information I am still skeptical about the significance of the weak and noisy CD signal.

We have now also recorded the CD spectrum of the apo ScATE1 and subtracted the apo protein spectrum, which is effectively a flat line in the visible region, from that of the [2Fe-2S]-bound data. This new spectrum is shown in Supplementary Fig. 3. Other than increasing the noise in the 600 nm – 800 nm region, the spectrum remains unchanged.

6. Thank you for the clarification and explanation. The 280 nm comparison shows how little FeS cluster is attached to this sample.

As we pointed out in points 2 and 3, we disagree with this assessment, and we believe our calculations based on the molar absorptivities (shown in point 3) and our iron and sulfide quantitations support significant cluster binding.

7. This data [inclusion of the mouse ATE1-1 spectral analyses] is a valuable addition and shows that the mouse and yeast proteins behave similarly. However, it also shows how little FeS is bound.

This comment is well taken, but as we discuss in point 3, we disagree with this assessment and show that this is not the case, albeit it there is some underloading for the [4Fe-4S] form of the protein, as rightfully pointed out by this reviewer.

8. Valuable information was added to some Figs. Thank you. (For Figs. S6 and S7: Why not simply diluting the samples and reading the 280 nm signal?).

For Figs. S6 and S7: those samples were generated anoxically and are frozen and stored at 77 K in EPR tubes. Thawing those samples could potentially result in their destabilizing and potential destruction; indeed, we tested such with the WT protein in EPR tubes, and the protein precipitates dramatically after a freeze-thaw cycle.

9. More important: Why is the 420 nm signal of the SxxS mutant (grey) so high in Fig. S10 (comparable to wild-type)? Why do the authors write: "The largest impact in cluster binding appears to alteration of the C20xxC23 motif to S20xxS23."? Mutation of the Cys should give NO signal at 420 nm, yet the signal is still there with additional absorption below 400 nm. Iron binding is usually at 320 nm. This Figure is confusing me.

We believe that the electronic absorption spectrum of the SxxS variant is completely different (merely a single, non-discrete absorption from 700 nm to 300 nm) from that of the WT protein (clear and distinct peak maxima centered around 405 nm; see above). Additionally, there is some Fe that binds to this variant (≈ 1 equivalent), but the binding motif is clearly different based on the major differences in the spectra. Perhaps the Fe is binding in the vicinal CC motif (maybe even in coordination with the now SxxS motif), but it is an artificial construct. Based on the altered spectrum and the drastically diminished metal stoichiometry, the protein's ability to bind a [4Fe-4S] cluster and its metal stoichiometry under these conditions are majorly compromised.

While idealized, it is also not true that absorption at 420 nm is completely abrogated when Cys residues are knocked out in all [Fe-S] proteins. As just one example, in the same *Science* paper referred to by this reviewer (Rouault, Krebs, Bollinger Jr. and coworkers *Science*, **373**, 236-241, (2021), DOI: 10.1126/science.abi5224), mutation of every ligating Cys residues still shows a broad absorption feature at 420 nm, the presence of some Fe binding, and no 320 nm feature. Moreover, as expected, we see a change in the yeast phenotype when any of these binding residues are mutated and expressed in our $\Delta ate1$ yeast strain, further confirming the functional relevance.

10. There are no new data added to address my concerns, even though the authors agree with my view that studying heme binding to proteins is a difficult issue.

We did not provide any additional data in our revision because it is well known in established literature that heme can interact adventitiously with proteins (see ref. 73 in our further revised manuscript Airola, M. V., Du, J., Dawson, J. H. & Crane, B. R. *Biochemistry* **49**, 4327-4338 (2010) as just one of several examples).

Importantly, the authors of the original *PNAS* manuscript that suggested ATE1 was a hemoprotein (Varshavsky and coworkers, *Proc. Natl. Acad. Sci. USA* **105**, 76-81 (2008)) have recently tested the binding of heme to *K. lactis* ATE1 (a homolog of *S. cerevisiae* ATE1), and they also report that the binding of heme causes aggregation and inactivation of *K. lactis* ATE1 (Varshavsky and coworkers *Proc. Natl. Acad. Sci. USA* **119**, e2209597119 (2022)). Thus, there is consensus in the ATE1 field that 1) the reaction of heme with ATE1 causes inactivation through aggregation; and 2) the interaction of heme with ATE1 is likely an off-target effect, as we also present here in this manuscript.

Reviewer 3

1. In comments to the editor the reviewer stated that the added work has improved the *in vitro* aspects of the work but the work has not moved beyond this and it was never demonstrated that ATE1 formed a complex with the proposed [Fe-S] either in yeast or mammalian cells if such clusters exist.

To respond to this comment, we have provided additional *in vivo* data that point to ATE1 as an [Fe-S] protein. We hope the reviewer agrees that, within the scope of this manuscript that, the combination of our *in vitro* data and *in vivo* data points to ATE1 as a [Fe-S] binding protein.

The focus of the manuscript is on yeast ATE1, which cannot be overexpressed to sufficient quantities for spectral analyses without causing the pro apoptotic phenotype, as we tried several times to do so (both in our previous revision and this subsequent revision). To work around this issue, we used a $\Delta ate1$ yeast strain, generating a number of Cys \rightarrow Ser mutants, which we showed *in vitro* to disrupt [Fe-S] cluster

binding, and testing the yeast homeostatic stress response, which showed dramatic perturbations when these Cys residues were lost. The inclusion of the mouse *in vitro* work is meant to show that, at the protein level, mouse ATE1 binds an [Fe-S] cluster and has similar spectroscopic properties to the yeast protein, allowing us to speculate that the two proteins function similarly.

Finally, we have added in additional *in vivo* data using mutant yeast deficient in [Fe-S] cluster biogenesis and/or delivery and, as expected, we see deficiencies in *in vivo* arginylation. We used two knockout yeasts, *yfh1* Δ and *met18* Δ , which are known to disrupt the ISC/CIA pathways (used to deliver [Fe-S] clusters to proteins *in vivo*) with differing mechanisms. By expressing an arginylation reporter protein inside these mutant yeasts, we found that the endogenous arginylation activity inside the *yfh1* Δ and *met18* Δ yeasts are significantly lower compared to the WT yeast strain, despite expressing similar amount of ATE1, consistent with our other data. Thus these additional results provide significant and compelling evidence further supporting our claim that [Fe-S] clusters are linked to post-translational arginylation. These new data are presented on pp. 19-22 of our further revised manuscript.

2. The reviewer points out that the work, in places, is an extension of work reported in REF 38.

We apologize, but we are unclear to the exact meaning of this comment. Ref. 38 (Hu, R.-G., Wang, H., Xia, Z. & Varshavsky, A. *Proc. Natl. Acad. Sci. USA* **105**, 76-81 (2008)) is from a completely different laboratory. In their work they suggested that heme interacts with Cys residue C71, 72 in mouse ATE1. In our work, the data indicates that yeast ATE1 is an [Fe-S] cluster-binding protein with Cys residues 20, 23 and also 94, 95, which correspond to Cys 71, 72 in mouse ATE1. As such our result has some implications to the previous work. However, our data show that heme binding to yeast ATE1 causes protein aggregation and enzymatic inactivation, which would be consistent to this previous work in *PNAS* showing that exogenous supplement of heme causes ATE1 destabilization. The previous results themselves do not actually indicate that ATE1 is a hemoprotein, only that heme can inactivate ATE1.

Our heme-binding and inactivation results are also fully consistent with a recent report from Varshavsky and coworkers that show that the binding of heme causes aggregation and inactivation of *K. lactis* ATE1 (Varshavsky and coworkers *Proc. Natl. Acad. Sci. USA* **119**, e2209597119 (2022)). Thus, with our data combined, it is clear that the reaction of heme with ATE1 causes inactivation through aggregation.

3. The reviewer also states that the evidence disagreeing with the literature view is limited and that evidence to support the literature view continues to be published.

We apologize, but without additional context, we have do not know how to interpret this comment because we are not sure which literature is of concern. In general, due to variations in experimental conditions and how models are tested, differences in conclusions are common in any field.

Reviewer 4

1. Regarding the position of the Cys motif in the different Ate1 isoforms, I was hoping for some insights into sequence alignment, not experiments. I understand that the current work is based on yeast ATE1, but at least some speculations would be helpful.

Our apologies for the misinterpretation of this request. A portion of this information is actually presented in the partial sequence alignments seen in Fig. S8. In short, it is unclear whether the difference in the 1A vs. 1B exon inclusion would lead to a difference in substrate specificity, as this alternative splicing leads to only a minor difference in the length (7 amino acids) of the extreme N-terminus of the protein. However,

we do not believe this difference will lead to a difference in cluster behavior. There is some difference in the flanking sequence of the CGYC motif, but the CGYC and CC motifs still align strongly across all eukaryotes, and the intervening length of the polypeptide between these regions only differs by a few amino acids in length. The 7A vs. 7B exon inclusion leads to changes in more remote regions of ATE1, and these could be more consequential in substrate specificity, although it is impossible to tell at this moment. We have added a couple of sentences of text in the Results immediately prior to the Discussion to address this point.

2. The manuscript is strangely formatted in my version, with many pages including only two lines or lines broken off in between. Some figure legends don't fit into the text boxes. I assume this will be corrected, but it made it more difficult to evaluate the revised manuscript.

We apologize for this issue, but this formatting problem was an error resulting from the manuscript tracking system, not from a formatting issue on our part. We have attempted to rectify this issue in our second resubmission.

3. I appreciate replacing the image in Fig. 2 with an Eppendorf tube to demonstrate the brownish color, but it would have been nice to have a comparison here -- an empty tube? A clear solution? something to illustrate the color difference?

We have now further modified Fig. 2 to show a comparison of the cluster-bound protein to the buffer control.

REVIEWERS' COMMENTS

Reviewer #2 (Remarks to the Author):

Reviewer 3

1. In comments to the editor the reviewer stated that the added work has improved the in vitro aspects of the work but the work has not moved beyond this and it was never demonstrated that ATE1 formed a complex with the proposed [Fe-S] either in yeast or mammalian cells if such clusters exist.

To respond to this comment, we have provided additional in vivo data that point to ATE1 as an [Fe-S] protein. We hope the reviewer agrees that, within the scope of this manuscript that, the combination of

our in vitro data and in vivo data points to ATE1 as a [Fe-S] binding protein.

The focus of the manuscript is on yeast ATE1, which cannot be overexpressed to sufficient quantities for

spectral analyses without causing the pro apoptotic phenotype, as we tried several times to do so (both

in our previous revision and this subsequent revision). To work around this issue, we used a Date1 yeast

strain, generating a number of Cys \rightarrow Ser mutants, which we showed in vitro to disrupt [Fe-S] clusterbinding, and testing the yeast homeostatic stress response, which showed dramatic perturbations when

these Cys residues were lost. The inclusion of the mouse in vitro work is meant to show that, at the protein level, mouse ATE1 binds an [Fe-S] cluster and has similar spectroscopic properties to the yeast

protein, allowing us to speculate that the two proteins function similarly.

Finally, we have added in additional in vivo data using mutant yeast deficient in [Fe-S] cluster biogenesis

and/or delivery and, as expected, we see deficiencies in in vivo arginylation. We used two knockout yeasts, yfh1D and met18D, which are known to disrupt the ISC/CIA pathways (used to deliver [Fe-S] clusters to proteins in vivo) with differing mechanisms. By expressing an arginylation reporter protein inside these mutant yeasts, we found that the endogenous arginylation activity inside the yfh1D and met18D yeasts are significantly lower compared to the WT yeast strain, despite expressing similar amount of ATE1, consistent with our other data. Thus these additional results provide significant and compelling evidence further supporting our claim that [Fe-S] clusters are linked to post-translational arginylation. These new data are presented on pp. 19-22 of our further revised manuscript.

ANSWER R2: A number of new additions to the manuscript has shown that ATE1 contains an [4Fe-4S] cluster. It is also shown that arginylation is compromised in ISC and CIA mutants. The Ser mutants of ATE1 are functionally hampered. These and other facts now clearly argue into the direction of ATE1 being an Fe/S protein. Overall, the new additions to the manuscript have made this claim much more convincing than in the previous versions.

2. The reviewer points out that the work, in places, is an extension of work reported in REF 38.

We apologize, but we are unclear to the exact meaning of this comment. Ref. 38 (Hu, R.-G., Wang, H.,

Xia, Z. & Varshavsky, A. Proc. Natl. Acad. Sci. USA 105, 76-81 (2008)) is from a completely different laboratory. In their work they suggested that heme interacts with Cys residue C71, 72 in mouse ATE1.

In our work, the data indicates that yeast ATE1 is an [Fe-S] cluster-binding protein with Cys residues 20,

23 and also 94, 95, which correspond to Cys 71, 72 in mouse ATE1. As such our result has some implications to the previous work. However, our data show that heme binding to yeast ATE1 causes protein aggregation and enzymatic inactivation, which would be consistent to this previous work in

PNAS

showing that exogenous supplement of heme causes ATE1 destabilization. The previous results themselves do not actually indicate that ATE1 is a hemoprotein, only that heme can inactivate ATE1. Our heme-binding and inactivation results are also fully consistent with a recent report from Varshavsky and coworkers that show that the binding of heme causes aggregation and inactivation of *K. lactis* ATE1 (Varshavsky and coworkers Proc. Natl. Acad. Sci. USA 119, e2209597119 (2022)). Thus, with our data combined, it is clear that the reaction of heme with ATE1 causes inactivation through aggregation.

ANSWER R2: The new work shows that the treatment with heme inactivates ATE1 by aggregation. This finding confirms the older papers. However, this interaction likely reflects an artefact rather than a functional contact (with, e.g., a regulatory role). Therefore, the previous papers may have missed the true function and cofactors of the protein. The new manuscript by no means is a simple extension of the Varshavsky papers. The new manuscript goes into an entirely new direction.

3. The reviewer also states that the evidence disagreeing with the literature view is limited and that evidence to support the literature view continues to be published. We apologize, but without additional context, we do not know how to interpret this comment because we are not sure which literature is of concern. In general, due to variations in experimental conditions and how models are tested, differences in conclusions are common in any field.

ANSWER R2: Admittedly, I also do not get this issue. Without further explanation by Rev 3, this concern is hard (impossible) to be addressed by the authors.

Comments for the re-revised version are marked with RERE.
Reviewer 2

1. Unfortunately, the authors have NOT responded to my major concern to analyze the in vivo relevance of FeS clusters on arginylation in yeast. In my eyes, a needed approach is the examination of ISC and/or CIA defects on arginylation to support the headline claim of the paper. Simply ignoring this point is confusing.

We respectfully accept this critique and apologize that we were unable to deliver these experiments in our previous revision. The disruption of ISC/CIA pathways is not easy because most of the involved enzymes are essential for yeast viability. Fortunately, we were able to identify two yeast strains with *yfh1* or *met18* knockouts, which are known to cause deficiencies in the ISC and/or CIA pathways via different mechanisms. We expressed an established arginylation reporter in these mutant yeasts to measure the in vivo arginylation levels. We found that the arginylation activity was indeed significantly reduced in these yeasts despite expressing similar amounts of ATE1 compared to the WT strain. Thus, these additional results provide significant and compelling evidence further strengthening our conclusion that [Fe-S] clusters are linked to post-translational arginylation. These new data are presented on pp. 19- 22 of our further revised manuscript.

RERE The activity of the arginylation assay is decreased in *Yfh1*- and *Met18*-deficient cells. This is an essential finding and a necessary addition to the manuscript.

2. I respectfully disagree here. Sulfide determination is simple, rapid (2 h) and does NOT need extensive amounts of material (at least in comparison the EXFAS, EPR and Moessbauer). It is a valuable (additional) measure to see whether Fe and S contents agree and whether close to stoichiometric amounts are bound to the protein.

We have regenerated the key WT ScATE1 samples in the manuscript and quantitated the sulfide amount according to a modified method of Beinert's (Anal. Biochem. 131, 373-378 (1983)) detailed in Ayikpoe,

R. et al. Biochemistry 58, 940-950 (2019). Briefly, cluster-containing protein was incubated 1% (w/v) zinc acetate and 7% (w/v) sodium hydroxide and incubated at room temperature for approximately 15 min. After incubation, 0.1% (w/v) N,N-dimethyl-p-phenylenediamine (dissolved in 5 M HCl) and 10 mM FeCl₃ (stabilized in 1 M HCl) were added, mixed, and the solution was incubated at room temperature for approximately 20 min. The chemically-generated methylene blue was measured at 670 nm with an extinction coefficient (ϵ_{670}) of $\approx 35 \text{ mmol L}^{-1} \text{ cm}^{-1}$, and the concentration of sulfide in the sample was determined.

As expected, we measured ~ 2 Fe per polypeptide and ~ 2 sulfide per polypeptide of the ScATE1 expressed in the presence of the ISC machinery, which binds a [2Fe-2S] cluster. As for the reconstituted protein, the sulfide is slightly sub-stoichiometric: we measured ~ 4 Fe per polypeptide and ~ 3 sulfide per polypeptide of the anoxically-reconstituted ScATE1. This latter result suggests some sub-stoichiometric loading of the [4Fe-4S] cluster, consistent with our slightly low molar absorptivity (see point 3 below). Nevertheless, our spectral data point to the presence of a [4Fe-4S] cluster. These new data are now presented in our further revised manuscript.

RERE Another essential addition to the manuscript. It shows that the possibly correct type of FeS cluster is a [4Fe-4S] species. This is important because it was not really clear previously what type of cluster the authors are dealing with.

3. For the current study I would have found this helpful, the more so as the spectral data indicate an under-stoichiometric binding of FeS (extinction coefficients are now extractable from the full spectrum UV/Vis data).

We respectfully disagree with this assessment. The yeast ATE1 A280 extinction coefficient is $\approx 136,000 \text{ M}^{-1} \text{ cm}^{-1}$ for the holo [4Fe-4S]₂⁺ protein based on amino acid analyses requested in the previous round of reviews by reviewer 1.

As just one example, we have graphed Fig. 3a below as molar absorptivity vs. wavelength. For the [4Fe-4S]₂⁺ species, ϵ at 405 nm (the peak maximum of the cluster) is $\approx 13,100 \text{ M}^{-1} \text{ cm}^{-1}$; for the [2Fe-2S]₂⁺ species, ϵ at 414 nm (the peak maximum of the cluster) is $\approx 19,700 \text{ M}^{-1} \text{ cm}^{-1}$.

These values are sufficiently in line with literature values, which can vary based on maximal wavelength reported; for example:

ϵ [4Fe-4S]₂⁺ FNR: $\approx 16,000 \text{ M}^{-1} \text{ cm}^{-1}$ (Thomson, LeBrun, and coworkers; see JACS, DOI: 10.1021/ja077455+)

ϵ [4Fe-4S]₂⁺ biotin synthase: $\approx 15,000 \text{ M}^{-1} \text{ cm}^{-1}$ (Jarrett and coworkers; see Biochemistry, DOI: 10.1021/bi0104625)

ϵ [4Fe-4S]₂⁺ WhiD: $\approx 16,000 \text{ M}^{-1} \text{ cm}^{-1}$ (Thomson, LeBrun, and coworkers; see Biochemistry, DOI: 10.1021/bi901498v)

ϵ [4Fe-4S]₂⁺ FeoC: $\approx 15,000 \text{ M}^{-1} \text{ cm}^{-1}$ (Smith and coworkers; see Biochemistry, DOI: 10.1021/acs.biochem.9b00745)

ϵ [4Fe-4S]₂⁺ NfuA: $\approx 15,000 \text{ M}^{-1} \text{ cm}^{-1}$ (Johnson and coworkers; see JBC, DOI: 10.1074/jbc.M709161200)

ϵ [2Fe-2S]₂⁺ RNR: $\approx 16,000 \text{ M}^{-1} \text{ cm}^{-1}$ (Fontecave and coworkers; see JACS, DOI: 10.1021/ja990073m)

ϵ [2Fe-2S]₂⁺ IscA: $\approx 10,000 \text{ M}^{-1} \text{ cm}^{-1}$ (Johnson and coworkers; see Biochemistry, DOI: 10.1021/bi3006658)

ϵ [2Fe-2S]²⁺ ferroxidase: $\approx 20,000 \text{ M}^{-1} \text{ cm}^{-1}$ (Johnson and coworkers; at $\approx 420 \text{ nm}$; see Biochemistry, DOI: 10.1021/bi00168a003) ϵ [2Fe-2S]²⁺ GLRX32-GS4: $\approx 17,000 \text{ M}^{-1} \text{ cm}^{-1}$ (Banci and coworkers; see JACS, DOI: 10.1021/jacs.0c02266)

Thus, while there is variability across the literature, our calculated molar absorptivity values indicate significant incorporation of the clusters. Our calculated molar absorptivity of our anoxically-reconstituted ScATE1 is slightly low compared to the average aggregate literature values, but as shown above there is some underloading of the replete [4Fe-4S] cluster based on sulfide quantitation.

RERE This quantitative analysis, together with the Fe and S determination, shows that the cluster binding site is almost fully occupied. Without these data, this could not be judged and data interpretation was difficult. Now, the analysis looks good.

4. As written above, I disagree in this general point, since sulfide determinations are quick and easy to do and do not need excessive amounts of material (at least in comparison to EXAFS and EPR requirements). I do not really understand why these simple additions were not provided (even though new mouse data were generated).

We have added the sulfide quantitation of the key samples, as described in point 2.

RERE OK.

5. The newly added insert for UV/Vis is a valuable and necessary information (and clearly shows the under-stoichiometric cluster contents). However, the correct control for the CD signal would be cluster-less protein not a buffer control, e.g., after destroying the cluster or a Cys mutant. Therefore, without new information I am still skeptical about the significance of the weak and noisy CD signal.

We have now also recorded the CD spectrum of the apo ScATE1 and subtracted the apo protein spectrum, which is effectively a flat line in the visible region, from that of the [2Fe-2S]-bound data. This new spectrum is shown in Supplementary Fig. 3. Other than increasing the noise in the 600 nm – 800 nm region, the spectrum remains unchanged.

RERE OK.

6. Thank you for the clarification and explanation. The 280 nm comparison shows how little FeS cluster is attached to this sample.

As we pointed out in points 2 and 3, we disagree with this assessment, and we believe our calculations based on the molar absorptivities (shown in point 3) and our iron and sulfide quantitations support significant cluster binding.

RERE See above, with the quantitation this issue seems OK now. Previously, the reader was left alone.

7. This data [inclusion of the mouse ATE1-1 spectral analyses] is a valuable addition and shows that the mouse and yeast proteins behave similarly. However, it also shows how little FeS is bound.

This comment is well taken, but as we discuss in point 3, we disagree with this assessment and show that this is not the case, albeit it there is some underloading for the [4Fe-4S] form of the protein, as rightfully pointed out by this reviewer.

RERE OK, as stated above.

8. Valuable information was added to some Figs. Thank you. (For Figs. S6 and S7: Why not simply diluting the samples and reading the 280 nm signal?).

For Figs. S6 and S7: those samples were generated anoxically and are frozen and stored at 77 K in

EPR tubes. Thawing those samples could potentially result in their destabilizing and potential destruction; indeed, we tested such with the WT protein in EPR tubes, and the protein precipitates dramatically after a freeze-thaw cycle.

RERE Too bad, but this can happen, unfortunately. Hence accepted, but it would have been an easy addition.

9. More important: Why is the 420 nm signal of the SxxS mutant (grey) so high in Fig. S10 (comparable to wild-type)? Why do the authors write: "The largest impact in cluster binding appears to alteration of the C20xxC23 motif to S20xxS23."? Mutation of the Cys should give NO signal at 420 nm, yet the signal is still there with additional absorption below 400 nm. Iron binding is usually at 320 nm. This Figure is confusing me.

We believe that the electronic absorption spectrum of the SxxS variant is completely different (merely a single, non-discrete absorption from 700 nm to 300 nm) from that of the WT protein (clear and distinct peak maxima centered around 405 nm; see above). Additionally, there is some Fe that binds to this variant (≈ 1 equivalent), but the binding motif is clearly different based on the major differences in the spectra. Perhaps the Fe is binding in the vicinal CC motif (maybe even in coordination with the now SxxS motif), but it is an artificial construct. Based on the altered spectrum and the drastically diminished metal stoichiometry, the protein's ability to bind a [4Fe-4S] cluster and its metal stoichiometry under these conditions are majorly compromised.

While idealized, it is also not true that absorption at 420 nm is completely abrogated when Cys residues are knocked out in all [Fe-S] proteins. As just one example, in the same Science paper referred to by this reviewer (Rouault, Krebs, Bollinger Jr. and coworkers Science, 373, 236-241, (2021), DOI: 10.1126/science.abi5224), mutation of every ligating Cys residues still shows a broad absorption feature at 420 nm, the presence of some Fe binding, and no 320 nm feature. Moreover, as expected, we see a change in the yeast phenotype when any of these binding residues are mutated and expressed in our

Δ ate1 yeast strain, further confirming the functional relevance.

RERE The comparison to the nsp12 should be cited correctly. When both!! FeS site are mutated in this Science paper, the residual absorbance is almost zero. Also the 55Fe binding is going down to zero (=background), when both!! sites are mutated. In contrast, the decrease in Supp Fig. 10 for the SxxS protein is only twofold. So, some FeS (or other species) binding still occurs. The protein seems to find weak alternative coordination sites ???

10. There are no new data added to address my concerns, even though the authors agree with my view that studying heme binding to proteins is a difficult issue.

We did not provide any additional data in our revision because it is well known in established literature that heme can interact adventitiously with proteins (see ref. 73 in our further revised manuscript Airola,

M. V., Du, J., Dawson, J. H. & Crane, B. R. Biochemistry 49, 4327-4338 (2010) as just one of several examples).

Importantly, the authors of the original PNAS manuscript that suggested ATE1 was a hemoprotein (Varshavsky and coworkers, Proc. Natl. Acad. Sci. USA 105, 76-81 (2008)) have recently tested the binding of heme to *K. lactis* ATE1 (a homolog of *S. cerevisiae* ATE1), and they also report that the binding of heme causes aggregation and inactivation of *K. lactis* ATE1 (Varshavsky and coworkers Proc. Natl. Acad. Sci. USA 119, e2209597119 (2022)). Thus, there is consensus in the ATE1 field that 1) the reaction of heme with ATE1 causes inactivation through aggregation; and 2) the interaction of heme with ATE1 is likely an off-target effect, as we also present here in this manuscript.

RERE OK, but this could have been stated more clearly for the readers, in my view.

We thank the reviewers for the further attentive and thoughtful suggestions (typed gray, below) to our manuscript. Responses to each reviewer (typed in black, below) are enumerated below.

Reviewer 2

9. More important: Why is the 420 nm signal of the SxxS mutant (grey) so high in Fig. S10 (comparable to wild-type)? Why do the authors write: "The largest impact in cluster binding appears to alteration of the C20xxC23 motif to S20xxS23."? Mutation of the Cys should give NO signal at 420 nm, yet the signal is still there with additional absorption below 400 nm. Iron binding is usually at 320 nm. This Figure is confusing me.

We believe that the electronic absorption spectrum of the SxxS variant is completely different (merely a single, non-discrete absorption from 700 nm to 300 nm) from that of the WT protein (clear and distinct peak maxima centered around 405 nm; see above). Additionally, there is some Fe that binds to this variant (≈ 1 equivalent), but the binding motif is clearly different based on the major differences in the spectra. Perhaps the Fe is binding in the vicinal CC motif (maybe even in coordination with the now SxxS motif), but it is an artificial construct. Based on the altered spectrum and the drastically diminished metal stoichiometry, the protein's ability to bind a [4Fe-4S] cluster and its metal stoichiometry under these conditions are majorly compromised.

While idealized, it is also not true that absorption at 420 nm is completely abrogated when Cys residues are knocked out in all [Fe-S] proteins. As just one example, in the same Science paper referred to by this reviewer (Rouault, Krebs, Bollinger Jr. and coworkers Science, 373, 236-241, (2021), DOI: 10.1126/science.abi5224), mutation of every ligating Cys residues still shows a broad absorption feature at 420 nm, the presence of some Fe binding, and no 320 nm feature. Moreover, as expected, we see a change in the yeast phenotype when any of these binding residues are mutated and expressed in our *ate1* yeast strain, further confirming the functional relevance.

RE RE The comparison to the *nsp12* should be cited correctly. When both!! FeS site are mutated in this Science paper, the residual absorbance is almost zero. Also the ^{55}Fe binding is going down to zero (=background), when both!! sites are mutated. In contrast, the decrease in Supp Fig. 10 for the SxxS protein is only twofold. So, some FeS (or other species) binding still occurs. The protein seems to find weak alternative coordination sites ???

We apologize if our reference to the aforementioned *nsp12* paper (DOI: 10.1126/science.abi5224) in our rebuttal letter were unclear. As this reviewer correctly points out, when all both binding sites of *nsp12* are mutated, there is nearly zero residue absorbance and zero additional binding of ^{55}Fe . As pointed out by this reviewer, we still see some modest Fe binding in the quadruple variant (approximately 1 mol eq.) pointing to a potentially adventitious binding site on ATE1 in the absence of the cluster-binding site. Indeed, there are 11 additional Cys residues on ScATE1 in addition to the cluster-binding residues that could participate in adventitious metal binding. However, our additional *in vitro* (N-terminal truncation) and *in vivo* data (stress assays) point squarely to these four Cys residues as the key residues for cluster binding.

To clarify this issue further, we have revised this section of the text on p. 17 to reflect the fact that there is potential adventitious metal binding when the key Cys residues are altered:

"In contrast, however, analysis of the $\text{S}^{94}\text{S}^{95}$ variant showed that iron was still capable of binding, albeit at a reduced capacity (2.19 ± 0.17 ; $n = 3$); moreover, the electronic absorption spectral features of the $[\text{4Fe-4S}]^{2+}$ were diminished, indicating either a different binding mode in this variant or adventitious iron binding (Supplementary Figure 10). Analysis of the quadruple variant ($\text{S}^{20}\text{xxS}^{23}$ and $\text{S}^{94}\text{S}^{95}$) demonstrated recapitulation of the spectral behavior of the $\text{S}^{94}\text{S}^{95}$ variant with only minimal iron binding (1.25 ± 0.07 ; $n = 3$), implying indeed that some adventitious iron may still bind to ScATE1 in the absence of the cluster-coordinating ligands (Supplementary Figure 10), which is possible as there 11 additional Cys residues in ScATE1 aside from the cluster-binding residues. Nevertheless, these data are consistent with very early

in vitro studies on ScATE1 that demonstrated an importance of these four strongly-conserved N-terminal Cys residues (Cys^{20/23/94/95}) as well as our own *in vivo* data (*vide infra*).”

We believe this textual edit captures the essence of the last point brought up by Reviewer 2.